# STYLEGUIDE: PREVENT CONTENT LEAKAGE USING NEGATIVE QUERY GUIDANCE

## ABSTRACT

In the domain of text-to-image generation, diffusion models have emerged as powerful tools. Recently, studies on visual prompting, where images are used as prompts, have enabled more precise control over style and content. However, existing methods often suffer from content leakage, where undesired elements from the visual style prompt are transferred along with the intended style (content leakage). To address this issue, we 1) extend classifier-free guidance (CFG) to utilize swapping self-attention and propose 2) negative visual query guidance (NVQG) to reduce the transfer of unwanted contents. NVQG employs negative score by intentionally simulating content leakage scenarios which swaps queries instead of key and values of self-attention layers from visual style prompts. This simple yet effective method significantly reduces content leakage. Furthermore, we provide careful solutions for using a real image as a visual style prompts and for image-to-image (I2I) tasks. Through extensive evaluation across various styles and text prompts, our method demonstrates superiority over existing approaches, reflecting the style of the references and ensuring that resulting images match the text prompts.

## 1 INTRODUCTION

Text-to-image diffusion models (T2I DMs) excel at synthesizing images that correspond to given text prompts (Rombach et al., 2022; Ramesh et al., 2021). However, relying solely on text prompts may not allow for precise control over the desired output. Even with highly detailed text prompts, controlling the exact style of the resulting images remains challenging ( Figure 1 (a) and (b)). Text prompts fail to specify precise style elements such as color, shading, line details, surface texture, or polygon density.

To address this issue, there has been significant research into using reference images as visual style prompts. These approaches include fine-tuning the diffusion model with a set of images containing the same theme (Ruiz et al., 2023; Kumari et al., 2023), learning new text embeddings (Gal et al., 2022; Han et al., 2023a; Avrahami et al., 2023), and adapting cross-attention modules to incorporate image features (Ye et al., 2023; Wang et al., 2023). However, these methods require costly training and often let the content from the visual style prompts leak to the result, i.e., content leakage (Sohn et al., 2023).

In contrast, training-free methods (Hertz et al., 2023; Chung et al., 2024b; Alaluf et al., 2024) swap features in the self-attention layer: the key and value from the visual style prompt replace the ones in the original process. Their motivation that the query carries the content, and the key-value carries the style allows promising performance for style reflection. However, this decomposition is not always satisfactory, leading to trade-off relationship between style and content (e.g., content leakage or poor style reflection). Moreover, when they tackle image-to-image (I2I, i.e., style transfer) where the content and style are specified by visual content/style prompts, the style from the content prompt leaks to the result along with the content from the style prompt leaking to the result as shown in Figure 12. Hence, we need more than naive sampling with query and key-value swapping for I2I. Appendix A further discusses related work.

In this study, we propose a method to more effectively extract the desired elements, whether style or content, from visual prompts and a text prompt. Our approach builds on classifier-free guidance

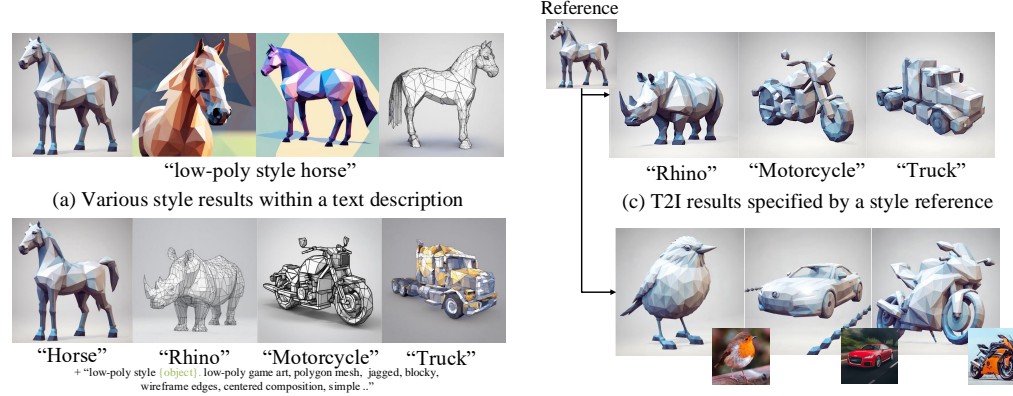

(a) Various style results within a text description

"low-poly style horse"

Reference

"Rhino"   "Motorcycle"   "Truck"

(c) T2I results specified by a style reference

"Horse"   "Rhino"   "Motorcycle"   "Truck"
+ "low-poly style [object]. low-poly game art, polygon mesh,  jagged, blocky,
wireframe edges, centered composition, simple .."

(b) Various style results with a highly detailed text description

(d) Style transfer results specified by a style reference

Figure 1: **Ambiguity of text prompts vs. visual style prompting.** (a) Ambiguity of text leads to different results within the same style description. (b) Even a detailed style description does not guarantee the generation of the same style images since it has many variants that can hardly be constrained using only text prompts. (c) Reference images can specify detailed visual elements.

(CFG) (Ho & Salimans, 2022) combined with swapping self-attention, enabling precise style transfer while maintaining the content specified by the text prompt. To address content leakage, we introduce negative visual query guidance, ensuring a clear separation between content and style. We also incorporate stochastic encoding for better style alignment and color calibration to match the final output to the reference image's color statistics.

We analyze where to apply swapping self-attention, identifying the optimal layers for balancing style transfer and content fidelity. Additionally, our method can effectively remove content that is difficult to eliminate with key and value swapping alone, working successfully even in cases with significant structural gaps between the style image and content text, as shown in Figure 3 (e.g., a complex scene of "a woman walking two dogs" and a single object "cat"). We extend the method to ControlNet for I2I style transfer, further enhancing its flexibility. Qualitative and quantitative evaluations show our method outperforms state-of-the-art approaches, providing precise control over content and style without content leakage. Our approach is both robust and efficient, ideal for visual style prompting tasks.[1]

## 2 VISUAL STYLE PROMPTING

We propose StyleGuide which receives a text prompt and a visual style prompt to generate new images. The results contain the content and style specified by the text prompt and the visual style prompt, respectively, with variations due to initial noises. The overview of our method is illustrated in Figure 2. First, we explain the swapping self-attention in the aspect of style transfer literature. StyleGuide consists of classifier-free guidance (CFG) with swapping self-attention, negative visual query guidance (NVQG), optimal layer choice, stochastic encoding of real visual style prompts, and generalization to ControlNet for real content images. We explain the first three components in text-to-image (T2I) scenario with generated visual style prompt. Then we proceed to T2I with real visual style prompt and image-to-image (I2I) scenario.

### 2.1 SWAPPING SELF-ATTENTION IN STYLE TRANSFER LITERATURE

Modern diffusion models consist of a number of self-attention and cross-attention blocks (Vaswani et al., 2017). Both of them employ the attention mechanism, which first obtains an attention map using similarity between query features $Q$ and key features $K$, then aggregates value features $V$ using the attention map as weights: $\text{Attention}(Q, K, V) = \text{Softmax}(\frac{QK^T}{\sqrt{d}})V$. Opposed to the cross-attention

---

[1]Our code will be released for reproducibility.

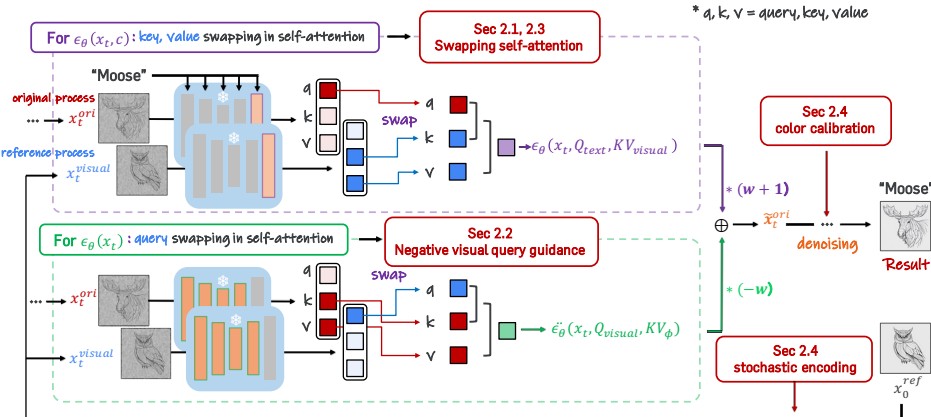

Figure 2: **Overveiw of StyleGuide**. Our proposed method includes 4 key components, highlighted in red boxes. First, stochastic encoding (Section 3.1) converts reference images into suitable latents for the visual style prompting task. Second, swapping self-attention (Section 2.2, 2.4) ensures the reference image's style is accurately reflected. Third, negative visual query guidance (Section 2.3) reduces content leakage from the reference image, allowing the desired text content (e.g., "Moose") to be better represented. Lastly, color calibration (Section 3.1) minimizes errors during the denoising process, helping to produce a cleaner final image.

layer, self-attention layer receives key and values coming from the main denoising process which has spatial dimensions with more freedom to represent spatially varying visual elements. As our goal is to reflect style elements from a reference image that are not easily represented by textual description, we opt to borrow key and values of self-attention layers in the reference process to the original process, namely swapping self-attention z (Figure 2). In addition, swapping self-attention has a strong connection with style transfer literature (Sheng et al., 2018; Park & Lee, 2019; Yao et al., 2019; Liu et al., 2021; Deng et al., 2022). where the attention mechanism reassembles visual features of a style image (key, value) on a content image (query).

## 2.2 CFG SAMPLING WITH SWAPPING SELF-ATTENTION FOR T2I

We propose CFG with swapping self-attention to reflect a visual style prompt in the T2I results. CFG (Ho & Salimans, 2022) is essential to guide the generated images toward given text prompts. For given a score $\epsilon_\theta(x_t, c)$ conditioned on $c$ and unconditional score $\epsilon_\theta(x_t, \emptyset)$, the CFG score is defined by $\tilde{\epsilon}_\theta = (1 + w)\epsilon_\theta(x_t, c) - w\epsilon_\theta(x_t, \emptyset)$ where $w$ controls the guidance strength. [2] CFG with $w > 1$ improves image quality and text alignment but excessive $w$ induces mode collapse (Chung et al., 2024a). Notably, CFG has not been explored in context of reflecting denoising process with modified features.

Assuming there exists the desired but hidden content $h^{\text{content}}$ and style $h^{\text{style}}$ of a given condition $c$, we formulate our task to model $p_\theta(x_0|h_{\text{text}}^{\text{content}}, h_{\text{visual}}^{\text{style}})$ using $\epsilon_\theta(x_t, c_{\text{text}})$ and $\epsilon_\theta(x_t, c_{\text{visual}})$ which are the original score leading to the original T2I-generated image $x_0^{\text{ori}} \sim p_\theta(x_0|c_{\text{text}}) = p_\theta(x_0|h_{\text{text}}^{\text{content}}, h_{\text{text}}^{\text{style}})$ and the reference score leading to the visual style prompt $x_0^{\text{visual}} \sim p_\theta(x_0|c_{\text{visual}}) = p_\theta(x_0|h_{\text{visual}}^{\text{content}}, h_{\text{visual}}^{\text{style}})$, respectively. We design the CFG score toward the result with the desired but hidden content $h^{\text{content}}$ and style $h^{\text{style}}$:

$$\tilde{\epsilon}_\theta(x_t, h_{\text{text}}^{\text{content}}, h_{\text{visual}}^{\text{style}}) = (1 + w)\ddot{\epsilon}_\theta(x_t, Q_{\text{text}}, KV_{\text{visual}}) - w\epsilon_\theta(x_t, \emptyset), \tag{1}$$

where $\ddot{\epsilon}_\theta(x_t, Q_{\text{text}}, KV_{\text{visual}})$ denotes a KV-injected denoising score as below. We use $\ddot{\epsilon}_\theta$ to indicate the score is not from a single condition but is estimated by feature manipulation.

For given two denoising processes, one as original and another as a reference, borrowing the key-value in self-attention from the reference to the original process, i.e., key-value (KV) injection, tends to

---

[2]We omit the diffusion timestep $t$ in the arguments and abuse $x_t$ instead of $z_t$ from latent diffusion model.

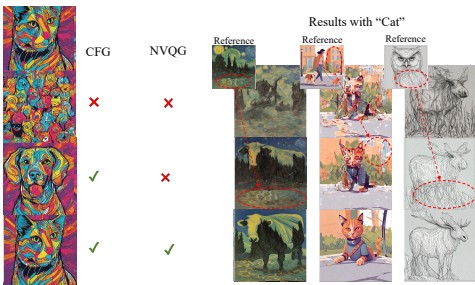

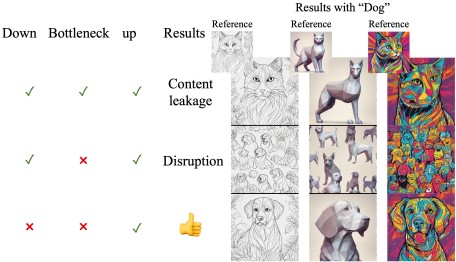

Figure 3: **The effect of CFG and the proposed negative visual query guidance on image generation.** The reference images provide the style for each generated output. Without NVQG, content leakage occurs, and the generated images fail to fully capture the intended content. In contrast, using NVQG ensures better alignment with both the reference style and the 'Cat' prompt, reducing content distortion and improving quality.

Figure 4: **The effect of swapping self-attention across different blocks.** Swapping self-attention on the bottleneck and downblocks causes content leakage, resulting in cat-like images despite a dog prompt, while swapping on downblocks disrupts resulting images. We only apply swapping self-attention in the upblocks to reflect style elements effectively.

produces results with the content from the original process and the style from the reference process with limited control (Alaluf et al., 2024; Chung et al., 2024b; Xu et al., 2023).

We define the KV-injected score by $\ddot{\epsilon}_\theta(x_t, Q_{\text{text}}, KV_{\text{visual}})$ where $Q_{\text{text}}$ and $KV_{\text{visual}}$ denote the query from the original score[3] $\epsilon_\theta(x_t, c_{\text{text}})$ and the key, value from the reference score $\epsilon_\theta(x_t, c_{\text{visual}})$. Then KV-injected-Attention becomes:

$$\text{Attention}(Q_{\text{text}}, K_{\text{visual}}, V_{\text{visual}}) = \text{Softmax}(\frac{Q_{\text{text}}K_{\text{visual}}^\mathsf{T}}{\sqrt{d}})V_{\text{visual}}. \tag{2}$$

We omit the layer index for brevity.

Naive denoising process with $\ddot{\epsilon}_\theta(x_t, Q_{\text{text}}, KV_{\text{visual}})$ provides limited control in generating images with content and style specified by a text $c_{\text{text}}$ and a visual style prompt, respectively. Moving forward, our CFG with swapping self-attention in Eq. (1) enjoys higher image quality and more accurate text alignment than the naive denoising process as in the original CFG for T2I generation. The results are deferred to Section 4.2.

## 2.3 NEGATIVE VISUAL QUERY GUIDANCE

We propose negative visual query guidance (NVQG) to further reduce the content $h_{\text{visual}}^{\text{content}}$ from the visual style prompt appearing in the results. Briefly, NVQG negates the CFG of a score $\ddot{\epsilon}(x_t, Q_{\text{visual}}, KV_{\text{text}})$.

In Liu et al. (2022), a complex text condition $c$ is factorized as a set of conditions $\{c_0, c_1, ...\}$ and Bayes' rule induces $p_\theta(x_t|c_0, c_1, ...) \propto \Pi \frac{p_\theta(x_t|c_i)}{p_\theta(x_t)}$. Then, the score of the complex text condition $c$ becomes $\epsilon_\theta(x_t, c) = \epsilon_\theta(x_t, \emptyset) + \Pi(\epsilon_\theta(x_t, c_i) - \epsilon_\theta(x_t, \emptyset))$. It allows reducing a specific concept $\tilde{c}$ with composition by negating its guidance with scale $w_{\text{neg}}$:

$$\epsilon_\theta(x_t, c, \text{not } \tilde{c}) = \epsilon_\theta(x_t, c) - w_{\text{neg}}(\epsilon_\theta(x_t, \tilde{c}) - \epsilon_\theta(x_t, \emptyset)) \tag{3}$$

Although we design $\ddot{\epsilon}_\theta(x_t, Q_{\text{text}}, KV_{\text{visual}})$ to predict the score toward $p_\theta(x_0|h_{\text{text}}^{\text{content}}, h_{\text{visual}}^{\text{style}})$, $\ddot{\epsilon}(x_t, Q_{\text{text}}, KV_{\text{visual}})$ still contain $h_{\text{visual}}^{\text{content}}$. Assuming a hidden factorization $KV_{\text{visual}} = \{KV_{\text{visual}}^{\text{content}}, KV_{\text{visual}}^{\text{style}}\}$, Bayes' rule induces

$$p_\theta(x_t|Q_{\text{text}}, KV_{\text{visual}}^{\text{style}}, KV_{\text{visual}}^{\text{content}}) \propto p_\theta(x_t)\frac{p_\theta(x_t|Q_{\text{text}}, KV_{\text{visual}}^{\text{style}})}{p_\theta(x_t)}\frac{p_\theta(x_t|Q_\emptyset, KV_{\text{visual}}^{\text{content}})}{p_\theta(x_t)}. \tag{4}$$

---

[3]The original score and its query within are recursively altered by KV injection along the denoising process.

Note that, $\epsilon_\theta(x_t) = \epsilon_\theta(x_t, Q_\emptyset, KV_\emptyset)$ where the source of the Q,K,V is $\emptyset$. Then, we get the desired conditional score of $\hat{c} = \{Q_{\text{text}}, KV_{\text{visual}}^{\text{style}}\}$:

$$\epsilon_\theta(x_t, \hat{c}) \leftarrow w_{\text{visual}}(\ddot{\epsilon}_\theta(x_t, Q_{\text{text}}, KV_{\text{visual}}) - \epsilon_\theta(x_t)) - w_{\text{content}}(\ddot{\epsilon}_\theta(x_t, Q_\emptyset, KV_{\text{visual}}^{\text{content}}) - \epsilon_\theta(x_t)) + \epsilon_\theta(x_t), \tag{5}$$

where $w_{\text{visual}}$ and $w_{\text{content}}$ sets the strength of each classifier. By borrowing the ability of query injection which successfully conveys content Tumanyan et al. (2023); Alaluf et al. (2024); Chung et al. (2024b); Xu et al. (2023), we approximate $\ddot{\epsilon}_\theta(x_t, Q_\emptyset, KV_{\text{visual}}^{\text{content}}) \approx \ddot{\epsilon}_\theta(x_t, Q_{\text{visual}}, KV_\emptyset)$ Lastly, we insert Eq. (5) into Eq. (1). Empirically, we find that replacing $\epsilon_\theta(x_t)$ to $\ddot{\epsilon}_\theta(x_t, Q_{\text{visual}})$ brings similar results. Finally, we can simply reformulate diffusion sampling with $w' = w_{\text{visual}}(w + 1)$:

$$\epsilon_\theta(x_t, \hat{c}) \leftarrow (w' + 1)(\ddot{\epsilon}_\theta(x_t, Q_{\text{text}}, KV_{\text{visual}}) - w'\ddot{\epsilon}_\theta(x_t, Q_{\text{visual}}) \tag{6}$$

Since we can highly relate the $w'\ddot{\epsilon}_\theta(x_t, Q_{\text{visual}})$ to the concept negation in equation 3 which guides the negation of concept with a scale $w_{\text{neg}}$, we named the query term as negative visual query guidance.

### 2.4 Choosing blocks for swapping self-attention

Here we explore the depth of the self-attention blocks to be swapped in the sense of granularity of visual elements. Modern architecture of diffusion models roughly consists of three sections in a sequence: downblocks, bottleneck blocks, and upblocks. Given that the bottleneck of diffusion models contains content elements of the image (Kwon et al., 2023; Jeong et al., 2024; Park et al., 2023), we opt not to apply swapping self-attention to bottleneck blocks to prevent transferring contents in a reference image. Figure 4 shows that not swapping the bottleneck blocks prevents content leaking from the reference image. However, the synthesized images show disrupted results with seriously scattered objects. Furthermore, while swapping self-attention implements the reassembling operation, simply applying to all self-attention layers exposes a content leakage problem, where the content of the reference image influences the resulting image, as shown in the first row of Figure 4. I.e., the results contain cats even though the prompts specify "a dog". We conjecture that this phenomenon happens because feature maps of downblocks have unclear content layout (Cao et al., 2023; Meng et al., 2024), so substituting features based on this inaccurate layout causes the disrupted results. To avoid injecting unnecessary features, we choose to swap the key and value of self-attention only in upblocks.

We note that Hertz et al. (2023) applies self-attention operation at the all blocks and suffers content leakage. The last row of Figure 4 shows the success of our strategy.

## 3 Real images as references

### 3.1 Real images as visual style prompts

So far, we have assumed a *generated* visual style prompt $x_0^{\text{visual}} \sim p_\theta(x_0|c_{\text{visual}})$. Here, we allow *real* visual style prompts by 1) stochastic encoding and 2) color calibration.

We propose stochastic encoding to obtain $x_t^{\text{visual}} \sim q(x_t|x_0^{\text{visual}})$ by adding a random noise on $x_0^{\text{visual}}$ following the forward process of DMs (Ho et al., 2020):

$$\epsilon_t \sim \mathcal{N}(0, I), \; x_t^{\text{visual}} = \sqrt{\alpha_t} \cdot x_0^{\text{visual}} + \sqrt{1 - \alpha_t} \cdot \epsilon_t \tag{7}$$

At each timestep, we samples $\epsilon_t$ to encode $x_t^{\text{visual}}$. It ensures that $x_t^{\text{visual}}$ lies on the learned trajectory and does not suffer from accumulative numerical error due to iterative process of DDIM inversion used by previous methods (Hertz et al., 2023; Chung et al., 2024b). Furthermore, it does not need to store the intermediate latents as opposed to DDPM inversion used by Alaluf et al. (2024).

Although stochastic encoding performs better than DDIM inversion, a subtle color discrepancy occurs between the resulting images and the visual style prompts. We introduce color calibration at $x_t^{\text{ori}}$ in the original process to match the statistics of predicted $x_0^{\text{ori}}$ to predicted $x_0^{\text{visual}}$. In Gatys (2015), distance between channel-wise statistics is employed as a style loss for style transfer. In Song et al. (2020), predicted $x_0$ ($= \frac{x_t^{\text{visual}} - \sqrt{1-\alpha_t} \cdot \epsilon_\theta(\hat{x}_t)}{\sqrt{\alpha_t}}$) allows to estimate $x_0$ with high probability at intermediate timesteps using deterministic sampling. Inheriting the advantage, we execute adain operation to match mean&std of predicted $x_0^{\text{ori}}$ with those of $x_0^{\text{visual}}$. It allows precise color calibration

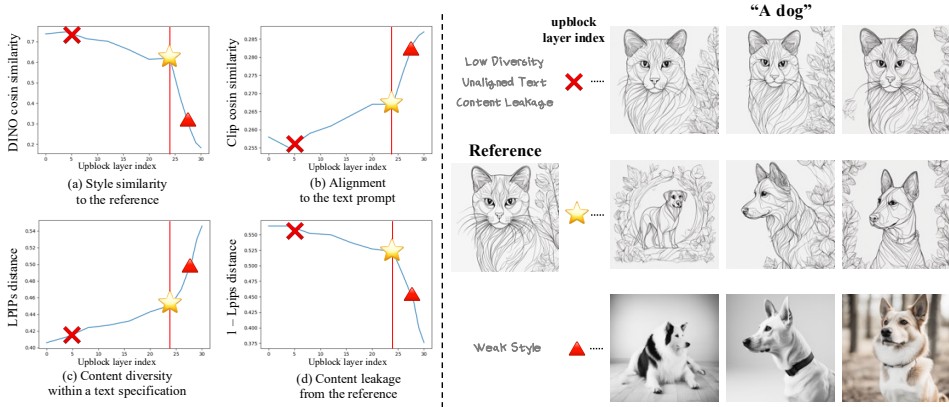

Figure 5: **Analysis on the optimal range of upblocks for swapping self-attention**. We find the optimal range of upblocks for a balanced trade-off between different aspects. The images on the right illustrate the visual results for different upblock layer indices, with the red cross indicating poor diversity and misalignment to the text prompt, the red triangle indicating a lack of style similarity, and the yellow star indicating the optimal results. Please refer to Section 2.4 for details.

rather than directly matching $x_t^{\text{ori}}$ to $x_t^{\text{visual}}$ in Alaluf et al. (2024); Chung et al. (2024b). Furthermore, ours differentiates from Chung et al. (2024b) in that using predicted $x_0$ at intermediate timestep $t \in (0, T)$ other than $x_T$ by reducing cumulative sampling error after the operation. Furthermore, Chung et al. (2024b) executes AdaIN at timestep T, inducing lengthy cumulative error.

We provide supportive experiments that show the effectiveness of the proposed method in Section 4.4 and a detailed Algorithm in Appendix B.2.

### 3.2 REAL IMAGE AS A CONTENT FOR STYLE TRANSFER

Our method can be extended not only to T2I tasks but also to I2I tasks, where users want to control the content using an image. In this I2I scenario, we adopt an approach where structural information from the content image is injected using ControlNet (Zhang et al., 2023).

Compared to our work, most existing self-attention variants (Chung et al., 2024b; Alaluf et al., 2024) for I2I style transfer employs query injection in self-attention to specify contents. However, the query from the content image contains not only the content elements (e.g., structure, layout, components) but also high nuance style elements (e.g., texture, pattern, and mesh) of the given image. As a result, style leakage can occur, transferring unwanted style elements from the content images. In the subsection 4.5, we demonstrate that our approach is more robust to style leakage issues compared to existing self-attention methods when dealing with real content images.

## 4 EXPERIMENTS

In this section, we describe the effects of our proposed methods: CFG with swapping self-attention, Negative visual query guidance (NVQG), stochastic encoding, and color calibration. For swapping self-attention, we provide a detailed analysis through experiments to determine the optimal layers for swapping. The impact of NVQGis demonstrated through qualitative results. Additionally, we show why stochastic encoding outperforms DDIM inversion when inverting real images, and we highlight the benefits of color calibration through experimental results.

We also conducted both quantitative and qualitative comparisons of our method against various competitors, including StyleAligned (Hertz et al., 2023), IP-Adapter (Ye et al., 2023), Dreambooth-LoRA Ruiz et al. (2023); Ryu (2023), StyleDrop (Sohn et al., 2023), DEADiff (Qi et al., 2024), CSGO (Xing et al., 2024) and InstantStyle (-plus) (Wang et al., 2024a;b). The details of these comparisons, along with the experimental setup and metrics, are described in the Appendix B.1.

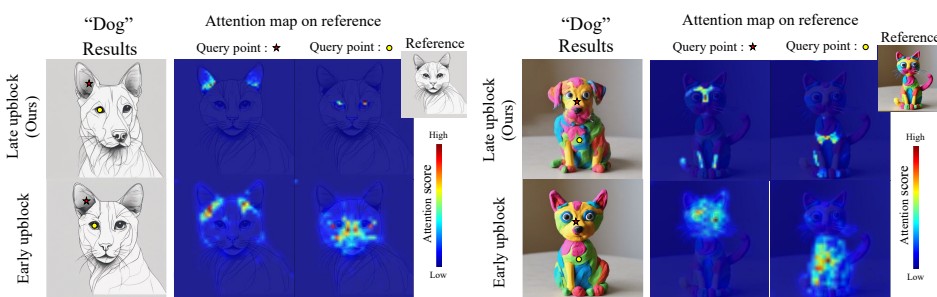

Figure 6: **Attention map visualization over late and early upblock layers.** The late upblock better focuses on the style-corresponding region than the early upblock, leading to more freedom to reassemble small parts. The early upblock attends larger region leading to content leakage.

## 4.1 ANALYSIS FOR SWAPPING SELF-ATTENTION

**Optimal layers in upblocks**   Since recent large T2I DMs consist of many layers, we further analyze the behavior by changing the start of the swapping while the end of the swapping is fixed at the end. We use four key metrics as shown in Figure 5, there is a layer where all four metrics abruptly change (red line). Notably, this point remains consistent regardless of the reference image. We choose this layer as the optimal start of the swapping for a balanced trade-off among all aspects. We provide qualitative results with detailed split of layers in Figure A2 and A3.

**Visualizing Attention maps**   Figure 6 compares average attention maps from the late upblock and the early upblock applying swapping self-attention. Using late upblock has more freedom to reassemble the reference style elements leading to more doggy results than early upblock which produces some cat-like attributes. The right two columns visualize the attention weight of query points marked as red stars and yellow dots. Swapping self-attention on late upblock reassembles features from a style correspondence, e.g., texture and color. On the other hand, swapping self-attention on early upblock reassembles features from a wider area with different styles. This comparison clarifies the reasons for using only late upblock. Please refer to Figure A5 for a detailed analysis.

## 4.2 EFFECTIVENESS OF CFG AND NVQG

This section analyzes the effects of Classifier-Free Guidance (CFG) and Negative Visual Query Guidance (NVQG) on image generation, with a focus on text alignment and content leakage. Figure 3 shows the results of the three configurations. **In the 1st row**, without CFG and NVQG, the generated images suffer from severe artifacts. The absence of CFG causes poor image quality resulting in significant misalignment with the prompt. **In the 2nd row**, CFG with swapping self-attention improves the text misalignment by boosting image quality. Here, the "cat" in target text prompt becomes clearer in the generated images. However, content leakage from the reference image still remains where unwanted elements (layouts, structure, and composition) from the reference affecting the results. **In the 3rd row**, NVQG releases the content leakage and produces the best results closely matching the text prompt while reflecting style reference.

Overall, Figure 3 demonstrates the critical role of NVQG in reducing content leakage, enjoying the quality boosting of CFG. Together, they ensure that the generated images align to both the target text prompt and the visual style prompts, resulting in high-quality, coherent outputs.

We qualitatively showcase diversity of results within a text prompt in Figure A7 and text alignment with complex text prompts in Figure A22.

## 4.3 COMPARISON AGAINST COMPETITORS

We compare ours with StyleAligned Hertz et al. (2023), IP-Adapter Ye et al. (2023), Dreambooth-LoRA Ruiz et al. (2023); Ryu (2023), and StyleDrop Sohn et al. (2023).

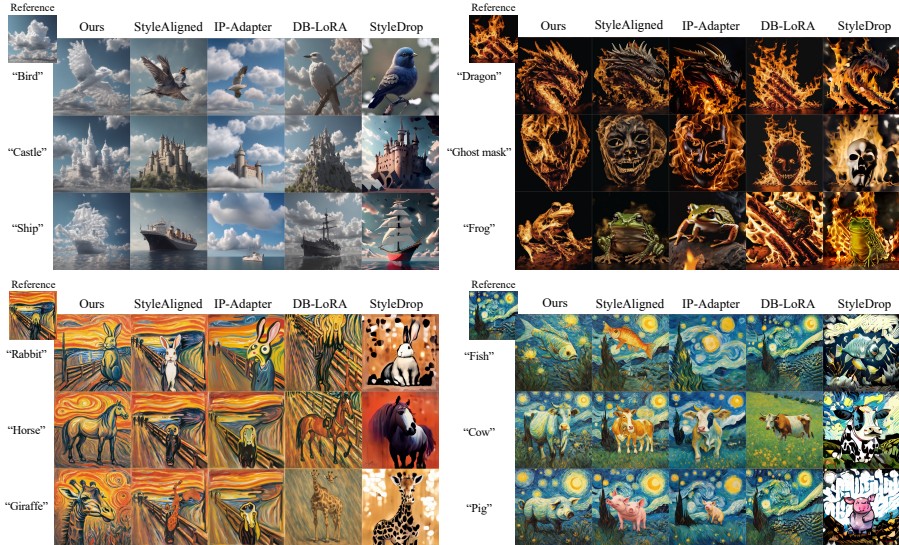

Figure 7: **Qualitative comparison across various styles and text prompts.** StyleGuide faithfully reflects style elements in reference images without causing content leakage from the reference images.

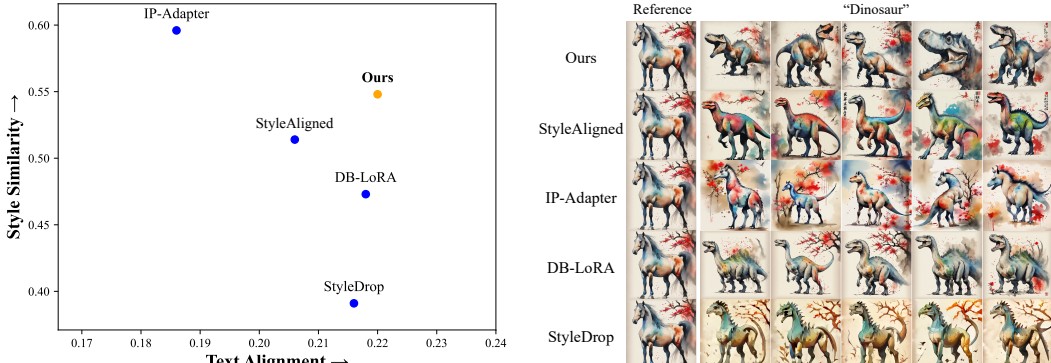

Figure 8: **Quantitative comparison**. We compare the results for text alignment (CLIP score) and style similarity (DINO embedding similarity) between other methods (blue points) and our method (orange point).

Figure 9: **Qualitative comparison with the same style.** Competitors face challenges in generating images with diverse layouts and compositions, i.e., content leakage from the reference.

**Style & content control** We provide a qualitative comparison in Figure 7, focusing on controlling style and content. Our method faithfully synthesizes content from the text prompt with the style of the reference image. In contrast, other methods add elements like color or texture not in the reference (e.g., feathers, bricks, iron, skin) and often suffer from content leakage (e.g., layout, screaming person, castle), which compromises text prompt faithfulness. Quantitative results in Figure 8 support these findings: IP-Adapter shows higher style similarity but neglects text prompts significantly. We provide additional comparisons with DEADiff (Qi et al., 2024), CSGO (Xing et al., 2024) and InstantStyle (-plus) (Wang et al., 2024a;b) in Figure A11.

**Diversity within a text specification** Starting from different initial noises, the diffusion models trained on a large dataset produce diverse results within the specification of a text prompt. Figure 9 shows that our results have various poses and viewpoints while others barely change, i.e., other methods limit the diversity of the pre-trained model. Figure A7 provides more examples.

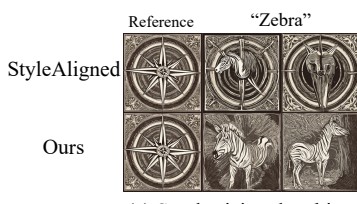
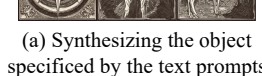
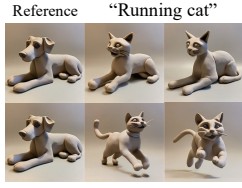
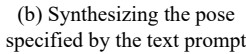
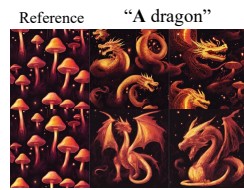

(a) Synthesizing the object specified by the text prompts

(b) Synthesizing the pose specified by the text prompt

(c) Synthesizing a single object specified by the text prompt

Figure 10: **Comparison for content leakage.** While StyleAligned suffers from content leakage from the reference, our results clearly align with the text prompts

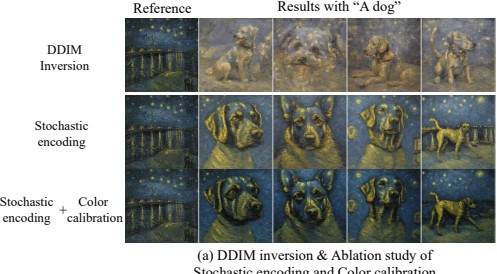
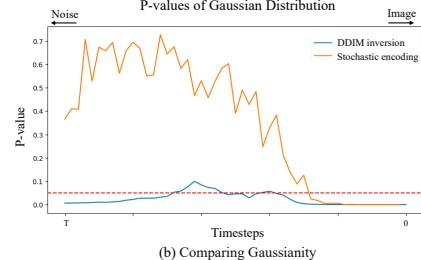

(a) DDIM inversion & Ablation study of Stochastic encoding and Color calibration

(b) Comparing Gaussianity

Figure 11: **(a) Comparison of DDIM inversion vs. stochastic encoding and the effect of color calibration.** Stochastic encoding reduces artifacts in the resulting images, while color calibration better reflects the colors of the reference image. **(b) Stochastic encoding produces the latents closer to the standard Gaussian distribution compared to DDIM inversion.** A P-value above 0.05 suggests that the data likely follows the standard Gaussian distribution.

**Content leakage**    Content leakage refers to the phenomenon where the content of a reference image appears in a result. As evident in Figure A10, our model exhibits significantly less content leakage when compared to other models. We focus on a comparison against a runner-up method, StyleAligned, particularly in terms of how content leakage can be an obstacle to controlling the content using a text prompt. Figure 10a compares examples where strong content leakage prevents the object from the text from appearing in StyleAligned. We often observe famous paintings in the reference easily leak into the results of StyleAligned while ours does not struggle. Figure 10b compares examples where the pose in the reference leaks into the results of StyleAligned preventing the reflection of specified pose in the text prompt. On the other hand, our method reflects the correct poses. Figure 10c compares examples where the number of small instances in the reference leaks into the result of StyleAligned. Contrarily, our method correctly synthesizes a single instance of the content specified by the text. In Figure A12, we provide quantitative comparison results. It supports that StyleGuide achieves the best style similarity while not suffering content leakage.

**Preserving the content of the original denoising process**    Figure A13 shows the results of ours and other methods using the same initial noise in each column. Our method precisely reflvects the style in the reference with minimal changes of contents in the original denoising process. On the other hand, the other methods severely alter pose, shape, or layouts. It is an important virtue of controlling style to keep the rest intact. We provide more results in Figure A14.

## 4.4 COMPARISON OF DDIM INVERSION WITH OUR STRATEGY

Figure 11 shows that StyleGuide can take real images as style reference with our strategies. As shown in Figure 11, stochastic encoding outperforms DDIM inversion and color calibration improves color consistency with the reference image. We provide more results in Figure A15. Moreover, we show that our strategy can boost the performance of the other self-attention variants Hertz et al. (2023); Cao et al. (2023) in Figure A16, A17 and color calibration can be used for generation settings in Figure A18. We provide an ablation study in Figure A19 for each configuration (swapping self-attention, NVQG, and

Figure 12: **Qualitative comparison in I2I style transfer task.** We compare our method for I2I style transfer task where the content image is given to control the content directly. Compared to the previous methods, our method transfer the reference style more accurately without style(e.g. color) leakage from the content image.

color calibration) in both real reference and generated reference settings. In both settings, swapping self-attention and NVQG improve style similarity and text alignment, while color calibration helps improve style similarity. Additionally, stochastic encoding demonstrates better performance than DDIM inversion.

## 4.5 COMPARISON IN STYLE TRANSFER

In Figure 12, we present a qualitative comparison between our method using ControlNet and existing state-of-the-art methods, CrossAttn (Alaluf et al., 2024) and StyleID (Chung et al., 2024b), for the I2I style transfer task where a content image is provided. Both CrossAttn and StyleID inject the query obtained by inversion from the content images. As discussed in subsection 3.2, the obtained query includes style elements from the content images, which results in the reference style not being properly reflected in the output. CrossAttn often fails to transfer the detailed representation of the reference style, leading to a rough, blocky appearance. For instance, when comparing the center examples in Figure 12, the style details are absent or inaccurately represented compared to our result. Similarly, StyleID struggles with style leakage, particularly with color information. In the center example, there is clear color leakage from the entire content image, and the same issue is visible in the rightmost example. In contrast, our method effectively reflects the details of the reference style image, with no noticeable transfer of the color values from the content images.

## 5 CONCLUSION AND LIMITATION

In this paper, we introduce StyleGuide using swapping self-attention, which effectively applies the style of reference images without content leakage in a training-free manner. **CFG with swapping self-attention** captures the reference image's style accurately and allows for direct content generation from the text, making it superior to other approaches. This integration with CFG enhances performance by balancing content generation and style transfer. To address content leakage in visual style prompting tasks, we propose **Negative visual guidance**, a simple method that ensures the reference image's content does not interfere with the text-specified content. Additionally, **Stochastic encoding** maps real images to suitable latents, improving the overall accuracy of the generated style, while **Color calibration** aligns the final output to the reference's color statistics.

Our method demonstrates qualitative and quantitative improvements over existing approaches, offering a robust solution for visual style prompting without complex training. We also provide a detailed analysis on where to apply swapping self-attention, identifying the optimal layers to balance style transfer and content fidelity. StyleGuide outperforms existing methods both qualitatively and quantitatively, and can be easily combined with algorithms such as ControlNet (Zhang et al., 2023) and Dreambooth-LoRA (Ryu, 2023), as shown in Figure A20.

However, StyleGuide is limited by the pretrained diffusion models' capabilities, unable to generate elements beyond the original model's scope (e.g., "stone golem" in Figure A24a). In adition the visually specified style overrides the textually specified style if they disagree as shown in Figure A24b.

For future work, expanding our method to other domains, such as video content, could broaden its applicability and open new research directions.

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
