# A RELATED WORK

## A.1 CONTROLLING STYLE WITH TRAINING

Dreambooth (Ruiz et al., 2023; Kumari et al., 2023; Everaert et al., 2023; Ryu, 2023) and Adapter (Zhang et al., 2023; Li et al., 2023; Sohn et al., 2023) variants fine-tune a pre-trained diffusion model (DM) with a few images with a same style to completely change the model. Textual inversion variants (Gal et al., 2022; Han et al., 2023a) learn a customized text embedding to use the same model with a minimal extra component. Meanwhile, ControlNet (Zhang et al., 2023) stands out as one of the most effective frameworks to guide structural contents. Similarly, adapter-based methods (Ye et al., 2023; Wang et al., 2023) attach an auxiliary image encoder which receives style reference. Several works(Šubrtová et al., 2023; Sun et al., 2023; Nguyen et al., 2023) train the model to solve image analogy where instructions demonstrate the desired manipulation.

Although these methods often succeed controling style, they require time-consuming additional training procedure or limits the applicable styles within its training set, which hinders practical usage.

## A.2 TRAINING-FREE CONTROL WITH FEATURE MANIPULATION

Manipulating the intermediate features in DMs inherently changes the resulting images even with a frozen DM.

Alaluf et al. (2024) and Chung et al. (2024b) transfer the style of an image to another image via self-attention with DDIM inversion achieving insufficient control. Furthermore, their main goal is image-to-image (I2I) rather than text-to-image (T2I); naive two-stage of T2I2I struggles in trade-off between style reflection and content preservation. StyleAligned (Hertz et al., 2023) similarly shares self-attention between two processes but fails to exclude the original style elements because it keeps the original features. Meanwhile, sharing self-attention features improves temporal appearance consistency over multiple frames (Wu et al., 2023; Yang et al., 2023) in video editing task.

Moving forward, we unveil the multiple missing components for accurate control over style and content, producing outstanding results.

On the other hand, Plug-n-Play (Tumanyan et al., 2023) and MasaCtrl (Cao et al., 2023) inject self-attention features from one process to another to convey *structure* and *object*, respectively, of the first process, rather than conveying style elements.

## A.3 INVERTING IMAGES TO NOISE FOR REAL IMAGE EDITING IN DIFFUSION MODELS

Training-free editing methods in the previous subsection manipulate the features of attention layers. If synthetic images with known latent noises are used as reference images, this process is straightforward. However, when editing with real images, the methods requires inverting the images into the initial latent noise maps. A majority of diffusion-based editing work (Alaluf et al., 2024; Hertz et al., 2022; Mokady et al., 2023; Couairon et al., 2022; Wallace et al., 2022) and self-attention variants (Hertz et al., 2023; Cao et al., 2023; Alaluf et al., 2024) employs DDIM inversion (Song et al., 2020) for its deterministic mapping between noise and image. However, DDIM inversion often suffers error at each step from $z_{t-1}$ to $z_t$ and inverted noise from a real image through DDIM does not follow the standard Gaussian distribution (Mokady et al., 2023; Miyake et al., 2023; Han et al., 2023b; Garibi et al., 2024).

DDPM sampling is an alternative that produces the standard Gaussian noise (Ho et al., 2020; Huberman-Spiegelglas et al., 2024; Wu & la Torre, 2023; Meng et al., 2022). However, they require lengthy iterative sampling for inversion or do not distinguish the reference denoising process from the inference denoising process. Compared to these inverting methods, our stochastic encoding consists of a one-step operation and produce statistically aligned intermediate latents.

## B   APPENDIX.

### B.1   EXPERIMENTS DETAILS AND METRICS

**Details**   We use SDXL Podell et al. (2023) as our pretrained text-to-image diffusion model and choose the $24^{th}$ layer and the after. We also validate our methods on Stable diffusion (SD) v1.5 Rombach et al. (2022). Results with SD v1.5 are shown in Figure A6. We set classifier-free guidance as 7.0 and run DDIM sampling with 50 timesteps following the typical setting. The initial noises for a text prompt are identical across competitors for fair comparison. We use the official implementation of IP-Adapter and StyleAligned. For Dreambooth-LoRA[4] , we use diffusers pipeline provided by Huggingface. Since the official code of StyleDrop is not available, we use an unofficial implementation[5] provided by Huggingface. All competitors are based on SDXL except StyleDrop (unofficial MUSE). As Dreambooth-LoRA, a training-based approach, requires multiple images, we train the models with five images: the original image and quarter-split patches of the reference image, because using only one image usually leads to destructive results or suffers from overfitting. For IP-adapter, we choose $\lambda = 0.5$ which is the best weighting factor for the task. For the visualization of the attention maps in Figure 6, we average the multi-head attention maps all together along the channel axis at the 20th denoising timestep.

**Metrics**   Following Voynov et al. (2023); Ruiz et al. (2023), we use DINO (ViT-B/8) embeddings Caron et al. (2021) to measure style similarity between a reference image and a resulting image. In addition, we provide a quantitative comparison with Gram matrix Gatys (2015) to assess style similarity in Table A1. We use CLIP (ViT-L/14) embeddings Radford et al. (2021) to measure the alignment between text prompts and resulting images. We use LPIPS Zhang et al. (2018) to measure diversity by average LPIPS between different resulting images in the same text prompt. We use Kolmogorov-Smirnov test Massey (1951) to measure gaussianity. For quantitative evaluation and comparison, we prepare 720 synthesized images from 40 reference images, 120 content text prompts (3 contents per 1 reference), and 6 initial noises. The reference images are generated from 40 stylish text prompts. Appendix B.3 provides the text prompt set. We also conducted a user study to evaluate the effectiveness of our method in Appendix B.4.

---

[4]https://huggingface.co/docs/diffusers/training/lora
[5]https://github.com/huggingface/diffusers/tree/main/examples/amused

Table A1: **Quantitative comparison.** We compare the results for style similarity by utilizing a gram matrix.

| Ours | StyleAligned | IP-Adapter | DB-LoRA | StyleDrop |
|---|---|---|---|---|
| **0.791** | 0.759 | 0.768 | 0.759 | 0.659 |

## B.2 Algorithm: Visual style prompting with a real image as a reference

---

**Algorithm 1:** Visual style prompting with a real image as a reference

---

**Input:** Reference latent $x_0^{visual}$, number of diffusion steps $T$, color calibration range $[t_{start}, t_{end}]$, precomputed constants $\alpha_t$, noise scale $\sigma_t$, model $\epsilon_\theta$

**Output:** Denoised latent $x_0$

1 **Function** `color_calibration`$(x_t, \hat{x}_t, x_0^{visual})$:

2    $x_{pred} \leftarrow \frac{x_t - \sqrt{1-\alpha_t} \cdot \epsilon_\theta(\hat{x}_t)}{\sqrt{\alpha_t}}$ ;                       `// predicted` $x_0$

3    $x_{dir} \leftarrow \sqrt{1-\alpha_{t-1}-\sigma_t^2} \cdot \epsilon_\theta(\hat{x}_t)$ ;        `// direction pointing to` $x_t$

4    $\epsilon \sim \mathcal{N}(0, I)$ ;                  `// Generate random noise`

5    $x_{noise} \leftarrow \sigma_t \cdot \epsilon$ ;                     `// random noise`

6    $\hat{x}_{pred} \leftarrow \text{adain}(x_{pred}, x_0^{visual})$ ;       `// calibrated predicted` $x_0$

7    $x_{t-1} \leftarrow \sqrt{\alpha_{t-1}} \cdot \hat{x}_{pred} + x_{dir} + x_{noise}$ ;         `// updated` $x_{t-1}$

8    **return** $x_{t-1}$

9 **Function** `adain`$(x, x^{visual})$:

10    $\mu_x, \sigma_x \leftarrow \text{channel\_wise\_mean\_std}(x)$ ;       `// mean and std of` $x$

11    $\mu_x^{visual}, \sigma_x^{visual} \leftarrow \text{channel\_wise\_mean\_std}(x^{visual})$ ; `// mean and std of` $x^{visual}$

12    $x_{norm} \leftarrow \frac{x - \mu_x}{\sigma_x}$ ;                   `// normalize` $x$

13    $x_{adain} \leftarrow \sigma_x^{visual} \cdot x_{norm} + \mu_x^{visual}$ ;        `// scale and shift`

14    **return** $x_{adain}$

15 **Function** `stochastic_encoding`$(x, t)$:

16    $\epsilon \sim \mathcal{N}(0, I)$ ;                  `// Generate random noise`

17    $x_t \leftarrow \sqrt{\alpha_t} \cdot x + \sqrt{1-\alpha_t} \cdot \epsilon$ ;

18    **return** $x_t$

19 Initialize $x_T \leftarrow \mathcal{N}(0, I)$;

20 Initialize $x_T^{visual} \leftarrow \text{stochastic\_encoding}(x_0^{visual}, T)$;

21 Initialize $t \leftarrow T$;

22 **for** $t = T$ **to** 1 **do**

23    $\hat{x}_t \leftarrow \text{CFG\_NQG\_with\_swapping\_self\_attention}(x_t, x_t^{visual})$;      `// (Fig 2) In denoising process,` $x_t$ `and` $x_t^{visual}$ `are swapped` ;                        `// and predict` $\hat{\epsilon}_t$

24    **if** $t_{start} \leq t \leq t_{end}$ **then**

25      $x_{t-1} \leftarrow \text{color\_calibration}(x_t, \hat{x}_t, x_0^{visual})$ ;

26    **end**

27    **else**

28      $x_{pred} \leftarrow \frac{\hat{x}_t - \sqrt{1-\alpha_t} \cdot \epsilon_\theta(\hat{x}_t)}{\sqrt{\alpha_t}}$ ;                   `// predicted` $x_0$

29      $x_{dir} \leftarrow \sqrt{1-\alpha_{t-1}-\sigma_t^2} \cdot \epsilon_\theta(\hat{x}_t)$ ;      `// direction pointing to` $x_t$

30      $\epsilon_t \sim \mathcal{N}(0, I)$ ;                `// Generate random noise`

31      $x_{noise} \leftarrow \sigma_t \cdot \epsilon_t$ ;                    `// random noise`

32      $x_{t-1} \leftarrow \sqrt{\alpha_{t-1}} \cdot x_{pred} + x_{dir} + x_{noise}$ ;        `// updated` $x_{t-1}$

33    **end**

34    Decrease $t$ by 1;

35 **end**

36 **return** $x_0$

---

### B.3 STYLE-CONTENT PROMPT LIST

1. the great wave off kanagawa in style of Hokusai: (1) book (2) cup (3) tree

2. fire photography, realistic, black background: (1) a dragon (2) a ghost mask (3) a bird

3. A house in stickers style.: (1) A temple (2) A dog (3) A lion

4. The persistence of memory in style of Salvador Dali: (1) table (2) ball (3) flower

5. pop Art style of A compass . bright colors, bold outlines, popular culture themes, ironic or kitsch: (1) A violin (2) A palm tree (3) A koala

6. A compass rose in woodcut prints style.: (1) A cactus (2) A zebra (3) A blizzard

7. A laptop in post-modern art style.: (1) A man playing soccer (2) A woman playing tennis (3) A rolling chair

8. A horse in colorful chinese ink paintings style: (1) A dinosaur (2) A panda (3) A tiger

9. A piano in abstract impressionism style.: (1) A villa (2) A snowboard (3) A rubber duck

10. A teapot in mosaic art style.: (1) A kangaroo (2) A skyscraper (3) A lighthouse

11. A robot in digital glitch arts style.: (1) A cupcake (2) A woman playing basketball (3) A sunflower

12. A football helmet in street art graffiti style.: (1) A playmobil (2) A truck (3) A watch

13. Teapot in cartoon line drawings style.: (1) Dragon toy (2) Skateboard (3) Storm cloud

14. A flower in melting golden 3D renderings style and black background: (1) A piano (2) A butterfly (3) A guitar

15. Slices of watermelon and clouds in the background in 3D renderings style.: (1) A fox (2) A bowl with cornflakes (3) A model of a truck

16. pointillism style of A cat . composed entirely of small, distinct dots of color, vibrant, highly detailed: (1) A lighthouse (2) A hot air balloon (3) A cityscape

17. the garden of earthly delights in style of Hieronymus Bosch: (1) key (2) ball (3) chair

18. Photography of a Cloud in the sky, realistic: (1) a bird (2) a castle (3) a ship

19. A mushroom in glowing style.: (1) An Elf (2) A dragon (3) A dwarf

20. The scream in Edvard Munch style: (1) A rabbit (2) a horse (3) a giraffe

21. the girl with a pearl earring in style of Johannes Vermeer: (1) door (2) pen (3) boat

22. A wild flower in bokeh photography style.: (1) A ladybug (2) An igloo in antarctica (3) A person running

23. low-poly style of A car . low-poly game art, polygon mesh, jagged, blocky, wireframe edges, centered composition: (1) A tank (2) A sofa (3) A ship

24. Kite surfing in fluid arts style.: (1) A pizza (2) A child doing homework (3) A person doing yoga

25. anime artwork of cat . anime style, key visual, vibrant, studio anime, highly detailed: (1) A lion (2) A chimpanzee (3) A penguin

26. A cactus in mixed media arts style.: (1) A shopping cart (2) A child playing with cubes (3) A camera

27. the kiss in style of Gustav Klimt: (1) shoe (2) cup (3) hat

28. Horseshoe in vector illustrations style.: (1) Vintage typewriter (2) Snail (3) Tornado

29. play-doh style of A dog . sculpture, clay art, centered composition, Claymation: (1) a deer (2) a cat (3) an wolf

30. A cute puppet in neo-futurism style.: (1) A glass of beer (2) A violin (3) A child playing with a kite

31. the birth of venus in style of Sandro Botticelli: (1) lamp (2) spoon (3) flower

32. line art drawing of an owl . professional, sleek, modern, minimalist, graphic, line art, vector graphics: (1) a Cheetah (2) a moose (3) a whale

33. origami style of Microscope . paper art, pleated paper, folded, origami art, pleats, cut and fold, centered composition: (1) Giraffe (2) Laptop (3) Rainbow

34. A crystal vase in vintage still life photography style.: (1) A pocket watch (2) A compass (3) A leather-bound journal

35. The Starry Night, Van Gogh: (1) A fish (2) A cow (3) A pig

36. A village in line drawings style.: (1) A building (2) A child running in the park (3) A racing car

37. A diver in celestial artworks style.: (1) Bowl of fruits (2) An astronaut (3) A carousel

38. A frisbee in abstract cubism style.: (1) A monkey (2) A snake (3) Skates

39. Flowers in watercolor paintings style.: (1) Golden Gate bridge (2) A chair (3) Trees (4) An airplane

40. A horse in medieval fantasy illustrations style.: (1) A castle (2) A cow (3) An old phone

### B.4 USER STUDY

For more rigorous evaluation, we conducted a user study with 62 participants. We configured a set with a reference image, a content text prompt, and six synthesized images with different initial noises per method from five competitors. The participants answered below question for 20 sets: Which method best reflects the style in the reference image AND the content in the text prompt? As indicated in Table A2, the majority of participants rated our method as the best. The lower ratings of the IP-Adapter in the user study may be attributed to its poor text alignment, as illustrated in Figure 8 despite its high style similarity. Examples of the user study are provided in Figure A1.

Please **select the option** that best satisfies both of the **following criteria**:
**1.The style** of the example image is reflected in the result image (e.g., color palette, texture, artistic style, material, etc.).
**2.The content** provided in the text matches the result image.

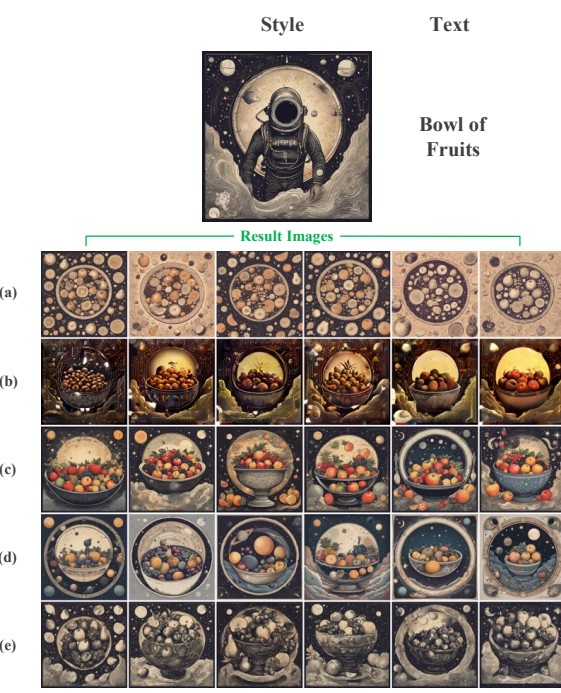

Figure A1: **Example of a user study.** Each row of images represents the result obtained by different method. The user had to assess which row is better in terms of style alignment and text alignment.

Table A2: **User study comparison.** We asked participants: Which method best reflects the style in the reference image AND the content in the text prompt?

| Ours | StyleAligned | IP-Adapter | DB-LoRA | StyleDrop |
|---|---|---|---|---|
| **58.15**% | 13.15% | 18.47% | 7.66% | 2.58% |

## B.5 ADDITIONAL FIGURE RESULTS

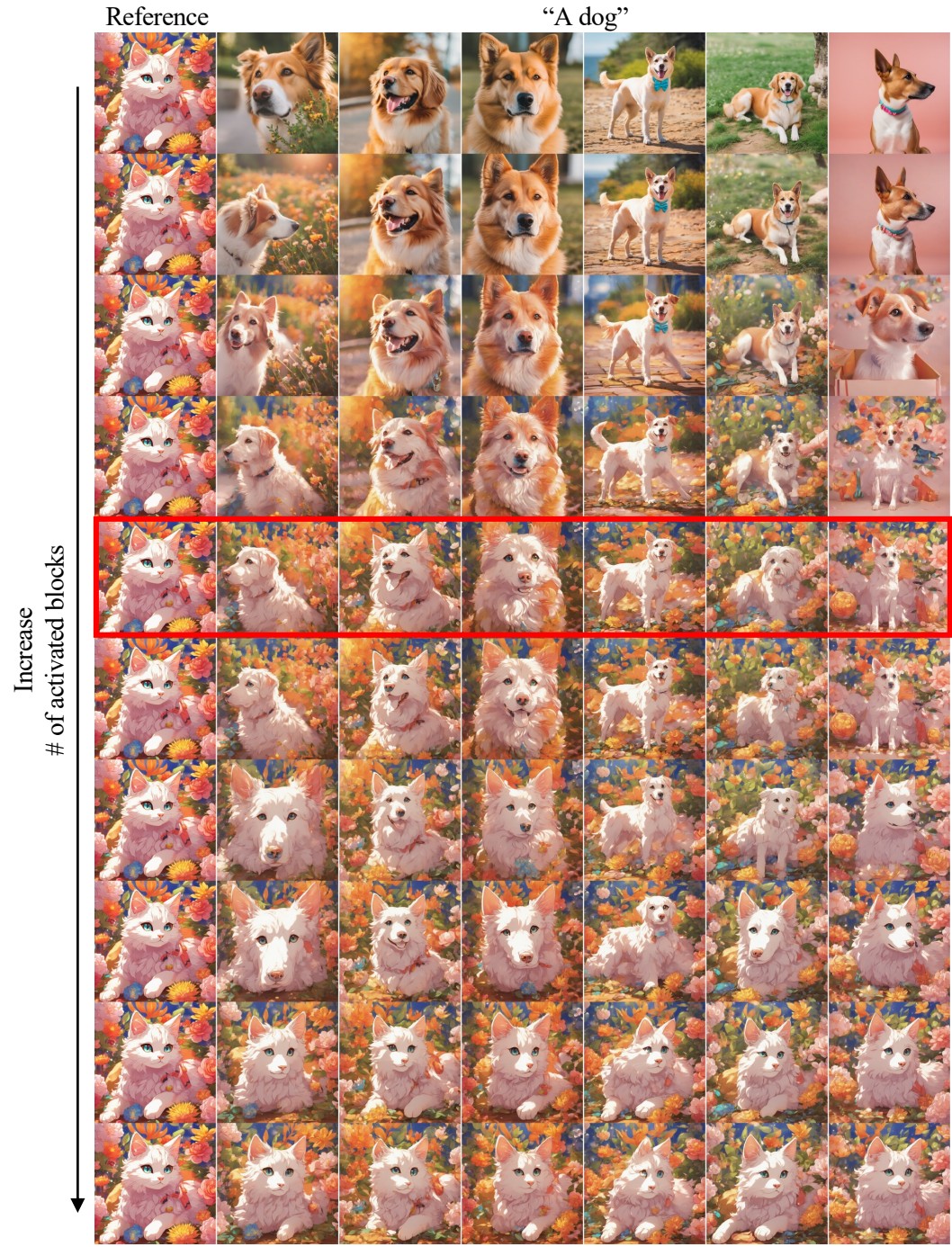

Figure A2: Selective cross style-attention is important to avoid content leakage while preserving style similarity. Content leakage decreases diversity and text alignment.

## B.6 MORE COMPARISONS WITH COMPETITORS

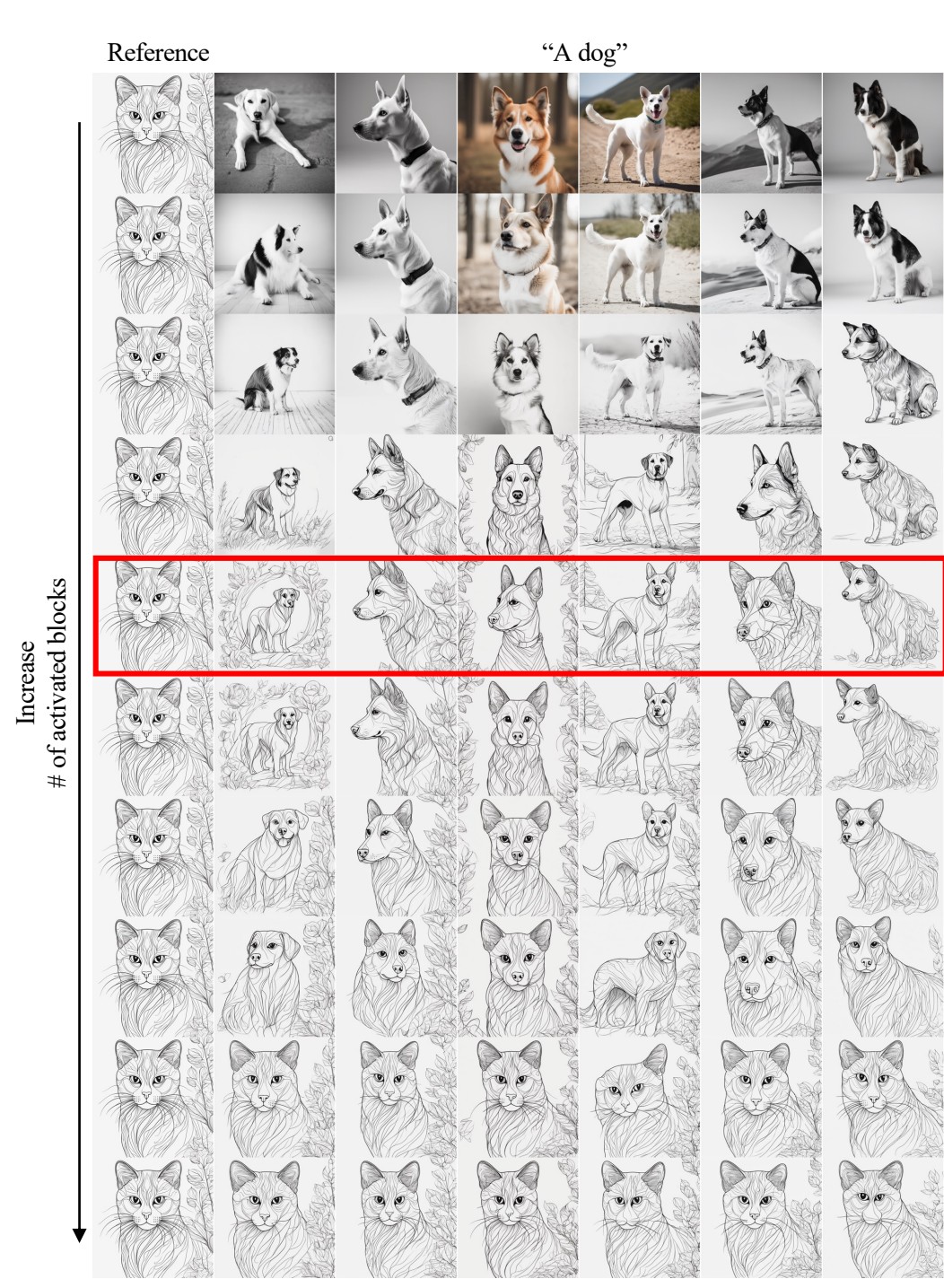

Figure A3: Selective cross style-attention is important to avoid content leakage while preserving style similarity. Content leakage decrease diversity and text alignment.

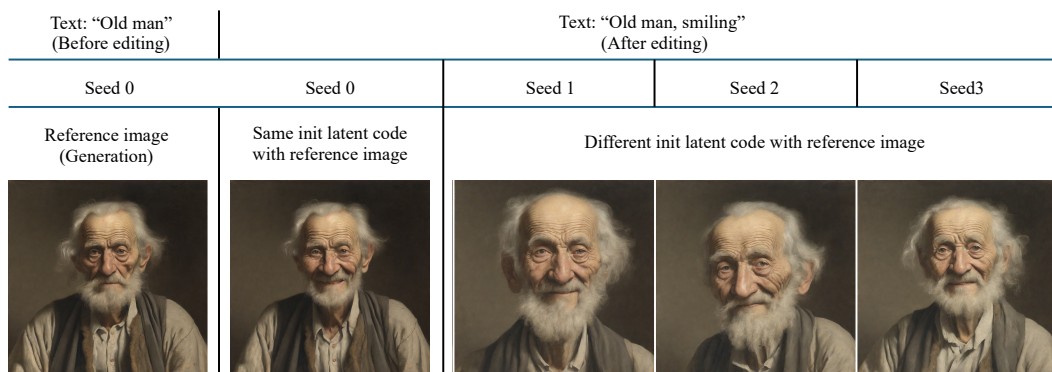

Figure A4: MasaCtrl Cao et al. (2023) employ the same initial latent for the reference and inference denoising process for preservation of the target identity. Different initial latent infer the different identities. In contrast, due to the differences in the tasks themselves, ours do not share the same initial latent between the reference denoising process and the inference denoising process.

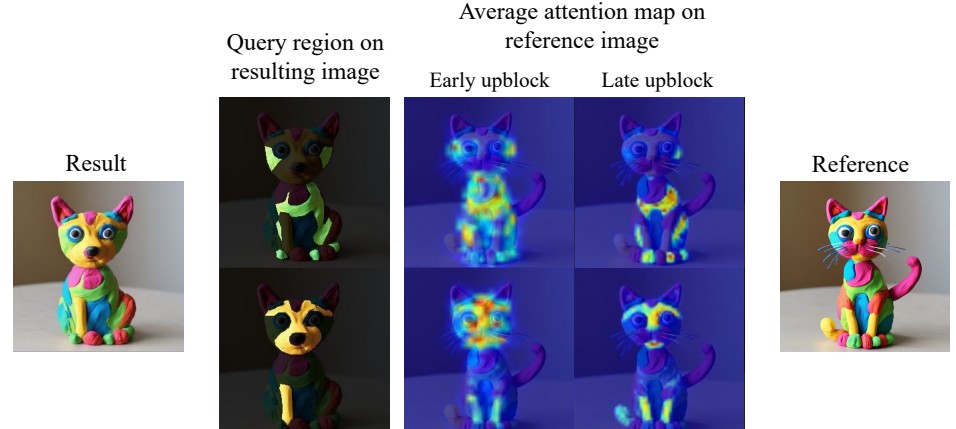

Figure A5: In Figure 6, we only show attention maps of 2 query points. Here, we provide the average attention map of multiple query points on the corresponding query region. At the late upblock, the query point region of the resulting image corresponds to the same style region of the reference image. On the other hand, at the early upblock, the query point region matches not only with the corresponding style region but also with the wider region

Reference                                            "House"

Reference                                            "Horse"

Reference                                            "Dog"

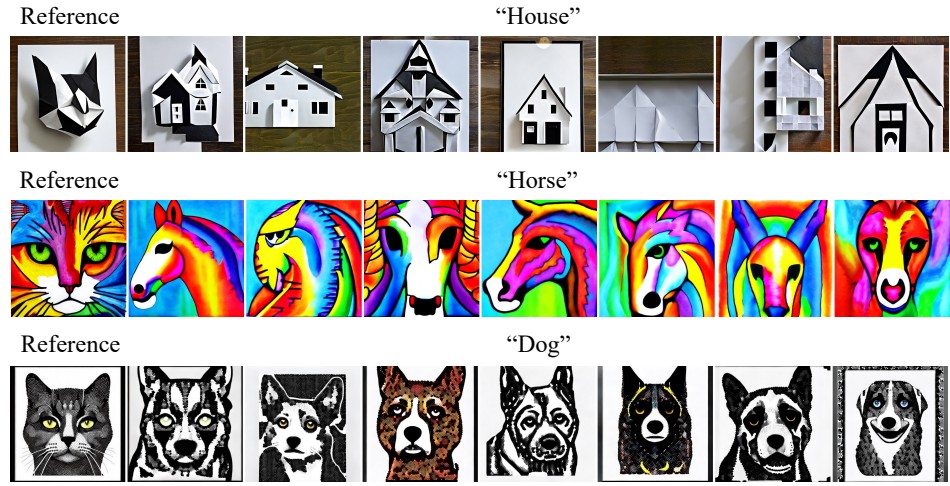

Figure A6: **Qualitative result of StyleGuide on stable diffusion v1.5** Ours also works on the other pretrained diffusion models.

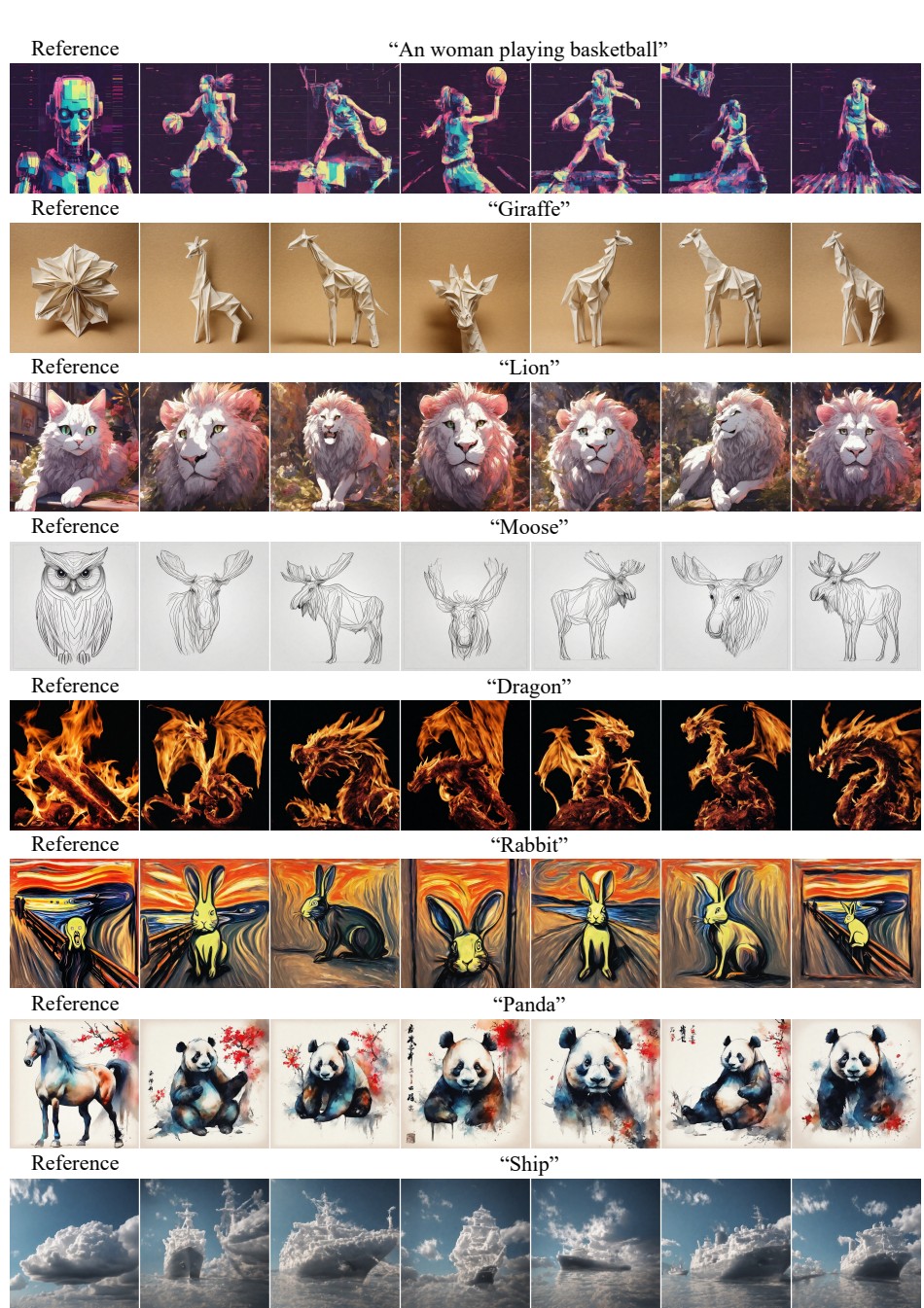

Figure A7: **Qualitative result of StyleGuide within a prompt.** Ours can generate diverse layouts, poses and composition within a prompt.

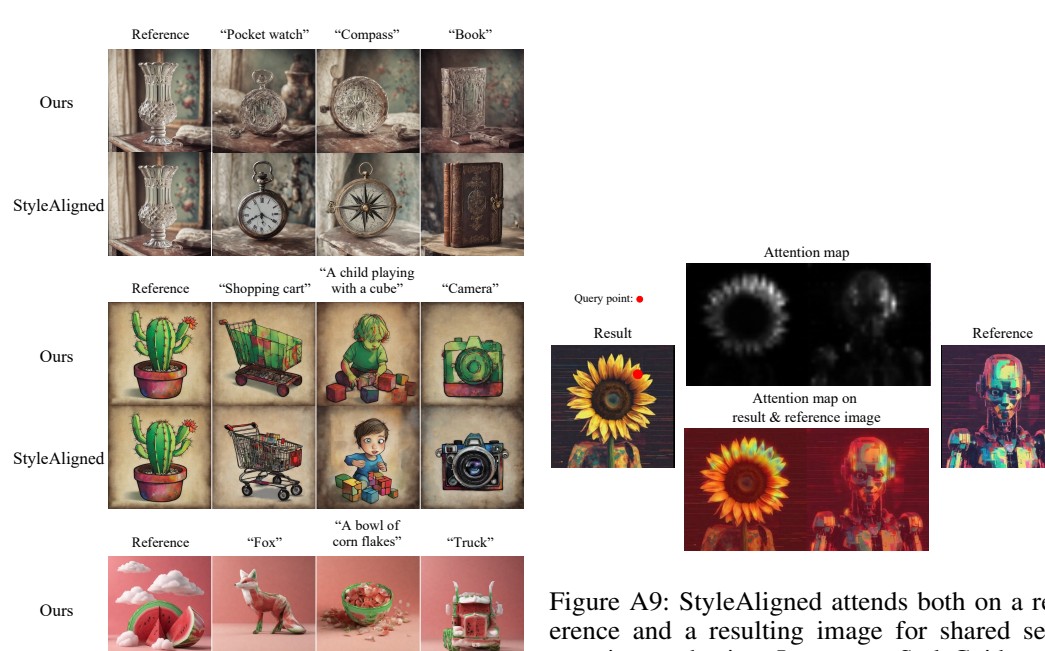

Figure A8: Definition of style is different between ours and StyleAligned.

Figure A9: StyleAligned attends both on a reference and a resulting image for shared self-attention mechanism. In contrast, StyleGuide only attends on a reference features which leads to better reflection of style in the reference image.

"Frog"

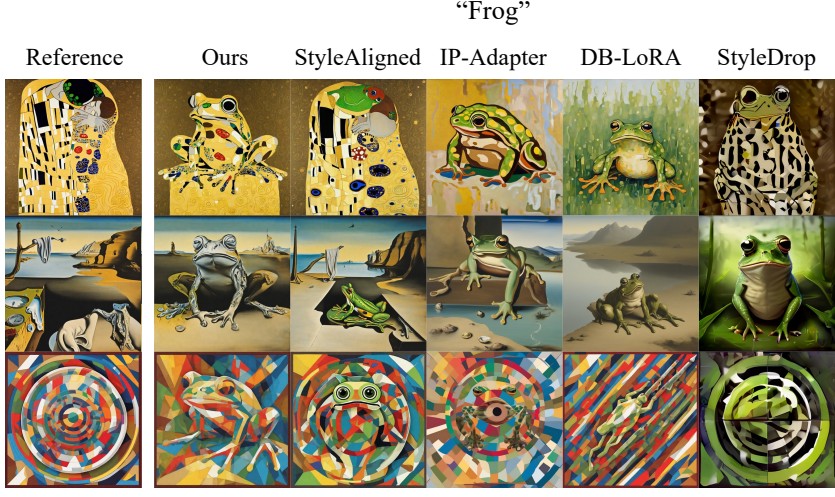

Figure A10: **Qualitative comparison with varying styles and fixed content.** StyleGuide reflects style elements from various reference images to render "frog" while others struggle.

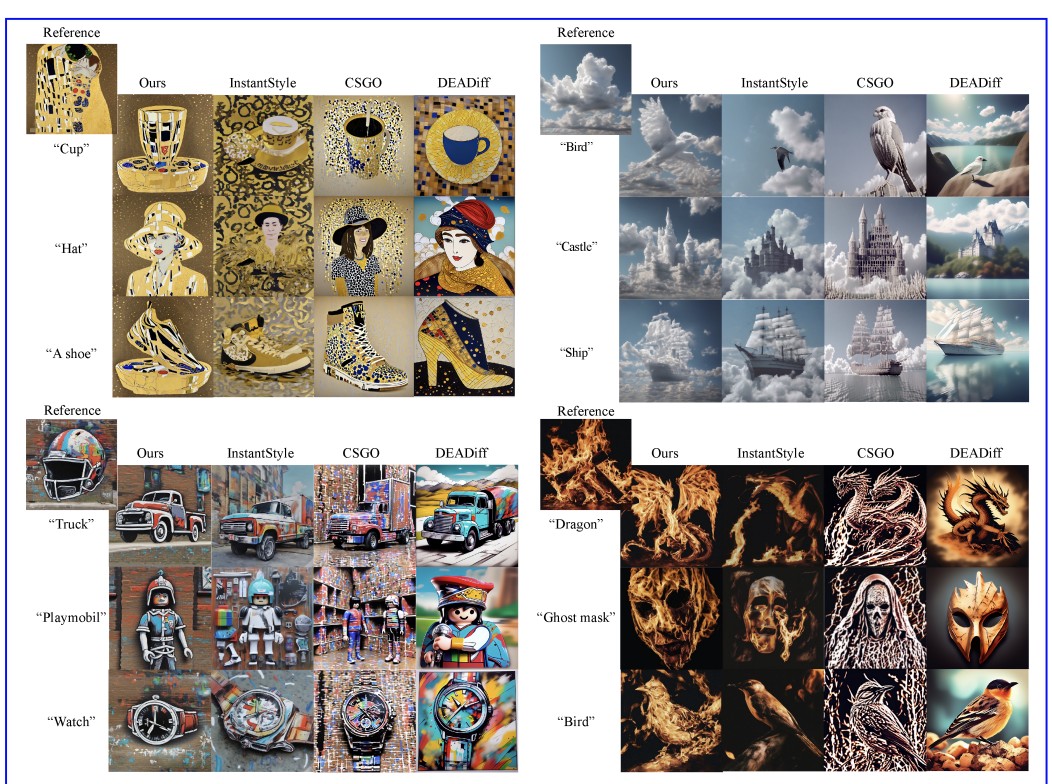

Figure A11: Our method shows better style reflection compared to the competitors

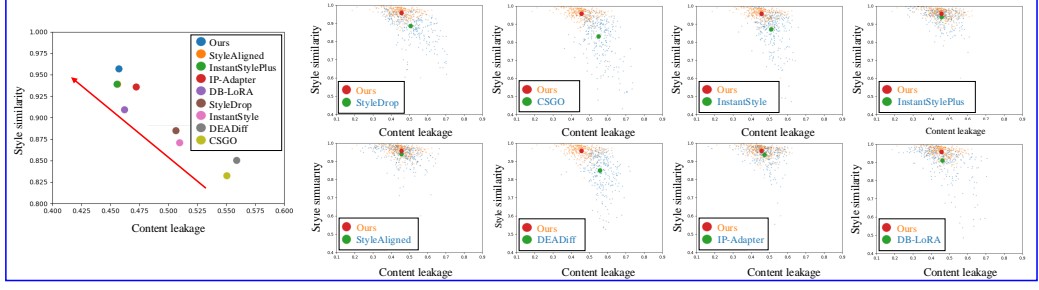

Figure A12: **Quantitative comparisons with content leakage and style similarity.** StyleGuide shows the best style similarity (Gram metric) while not suffering content leakage.

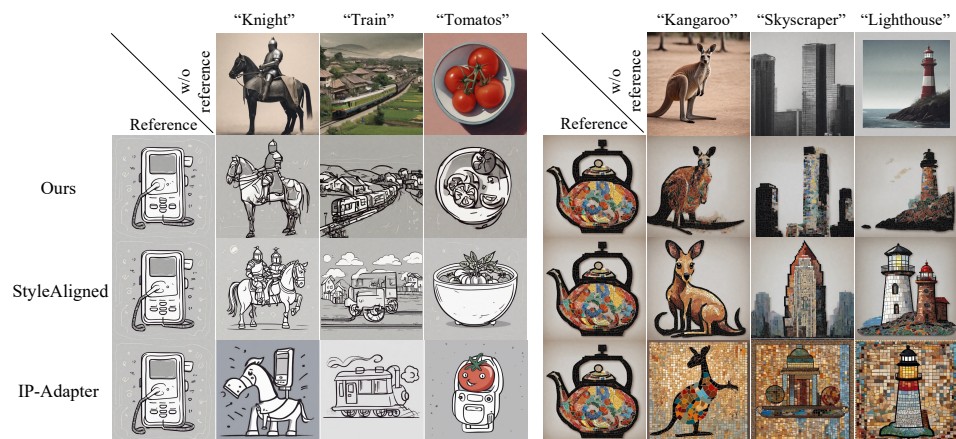

Figure A13: **Comparison of contents change while reflecting the style in the reference.** Each column shares the same initial noise. StyleGuide reflects the style in the reference with minimal changes in the contents of the original denoising process while others produce drastic changes.

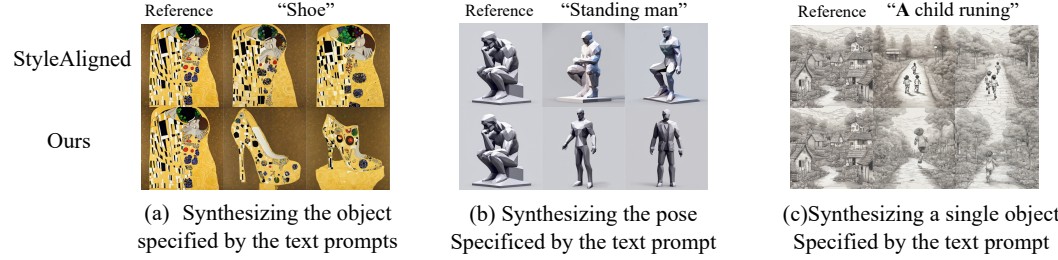

Figure A14: **Comparison for content leakage.** While StyleAligned suffers from content leakage from the reference, results from ours clearly align with the text prompts

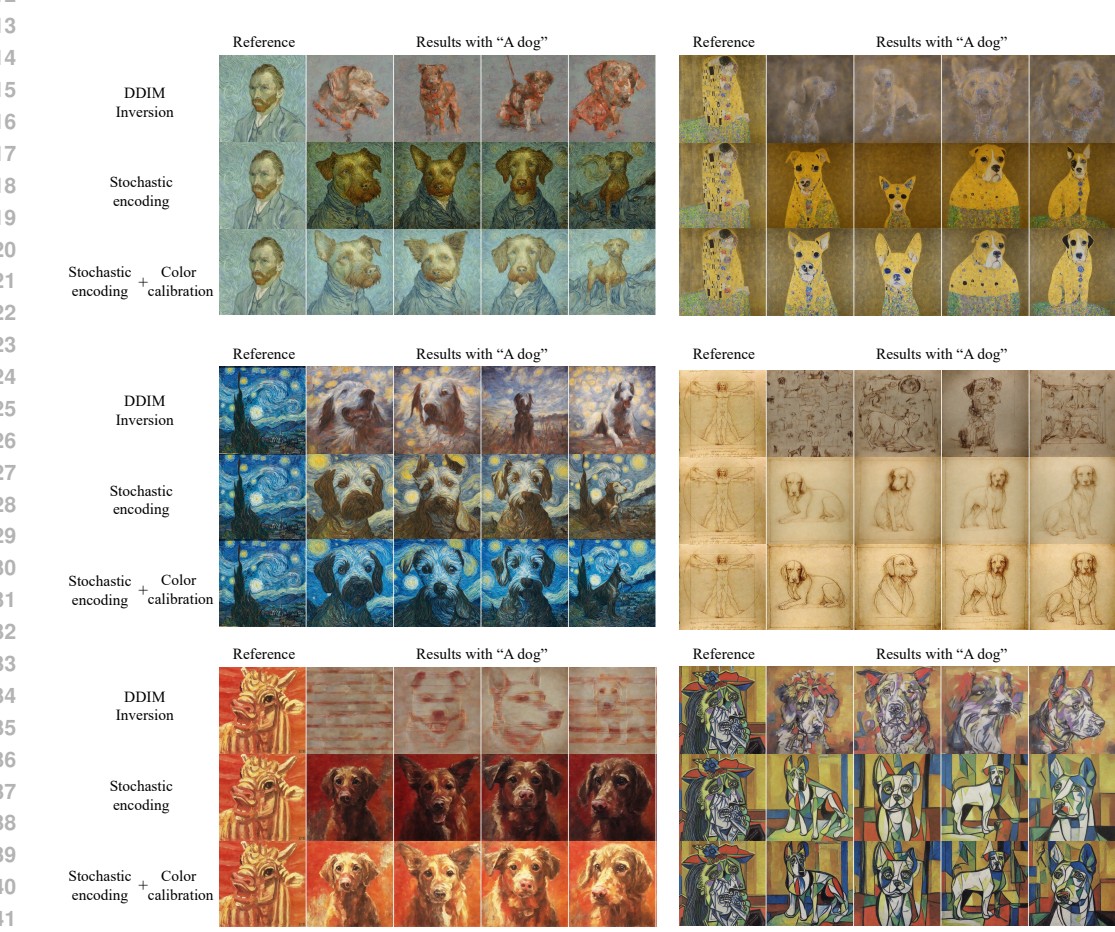

Figure A15: More comparison of DDIM inversion with stochastic encoding, and with stochastic encoding and color calibration. Stochastic encoding reduces artifacts in the generated images, while color calibration better preserves the colors of the reference image.

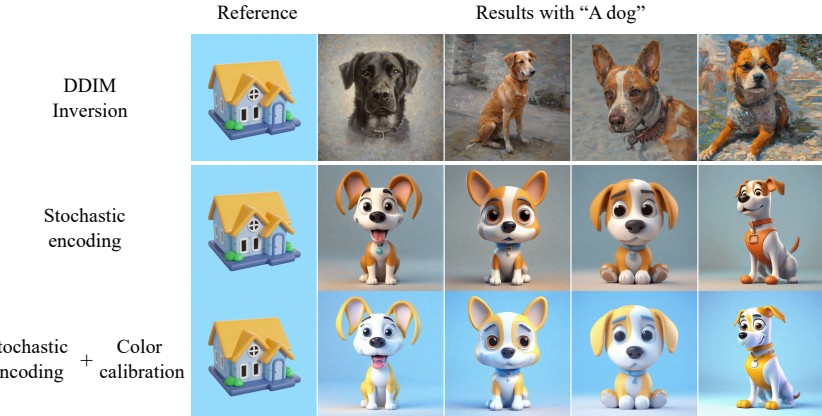

Figure A16: **Ablation study with StyleAligned Hertz et al. (2023).** Our strategy for a real reference image is compatible with self-attention variants for boosting the reflection of style elements.

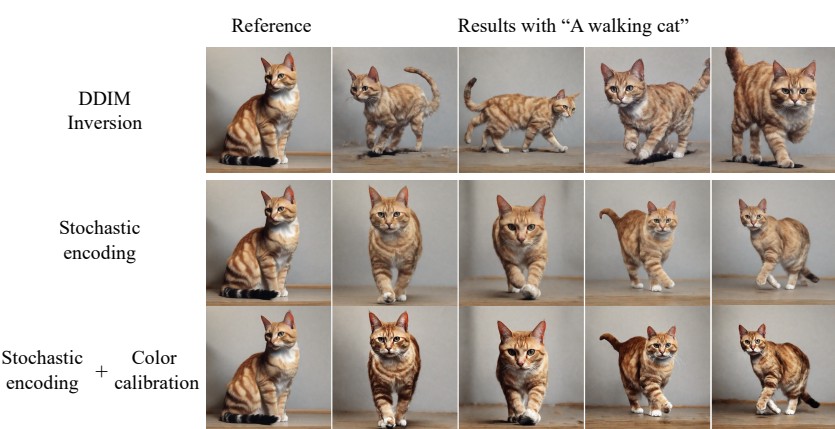

Figure A17: **Ablation study with MasaCtrl Cao et al. (2023).** Our strategy for a real reference image is compatible with self-attention variants for reducing artifacts and boosting the reflection of visual elements.

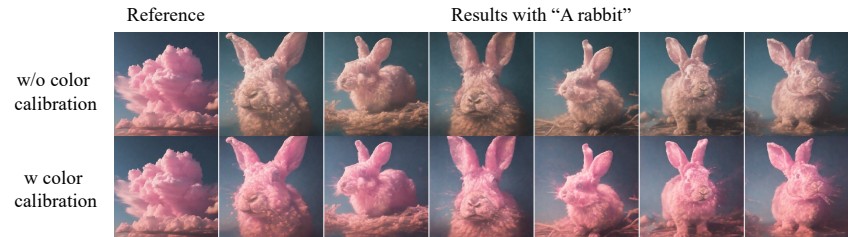

Figure A18: **Color calibration ablation study with generating process from random noise.**. Color calibration also decreases the minor discrepancy of color between the reference image and the resulting image that barely occurs in the generation setting.

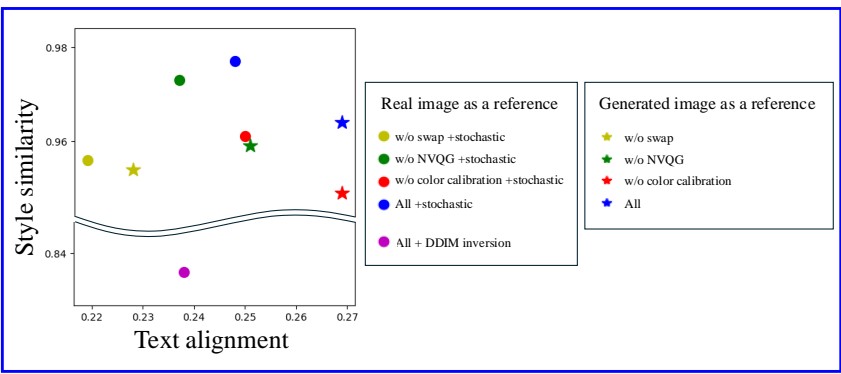

Figure A19: **Ablation study of four proposed methods with a real reference and a generated reference setting.**. For both settings, swapping self-attention and NVQG improve style similarity and text alignment. Color calibration helps to improve style similarity. Stochastic encoding shows better performance than DDIM inversion.

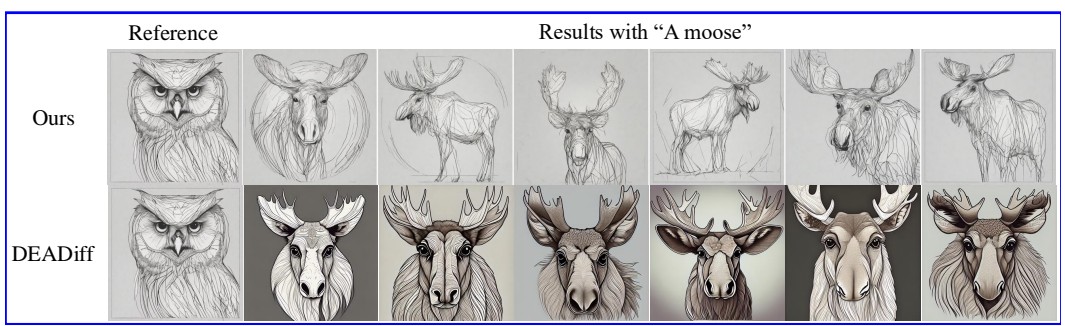

Figure A20: **Content leakage in DEADiff Qi et al. (2024)** The results of DEADiff suffer from content leakage (e.g., the frontal face of "A moose" and the "An owl" reference image), which reduces diversity within a text prompt. StyleGuide do not suffer content leakage while reflecting style.

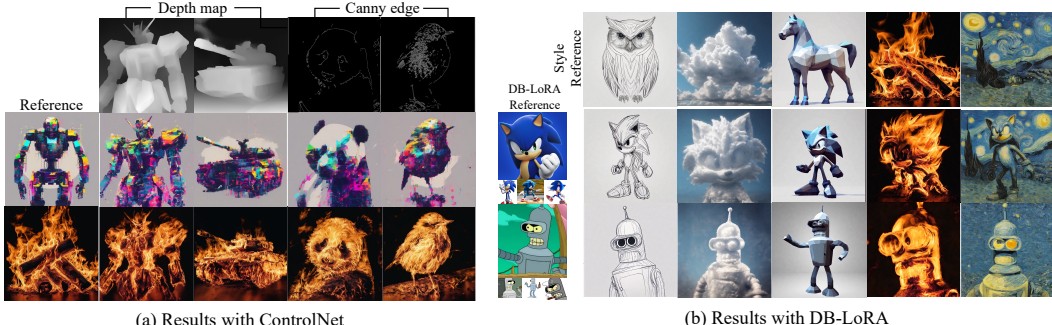

(a) Results with ControlNet        (b) Results with DB-LoRA

Figure A21: **Visual style prompting with existing techniques.** Our method is compatible with ControlNet and Dreambooth-LoRA.

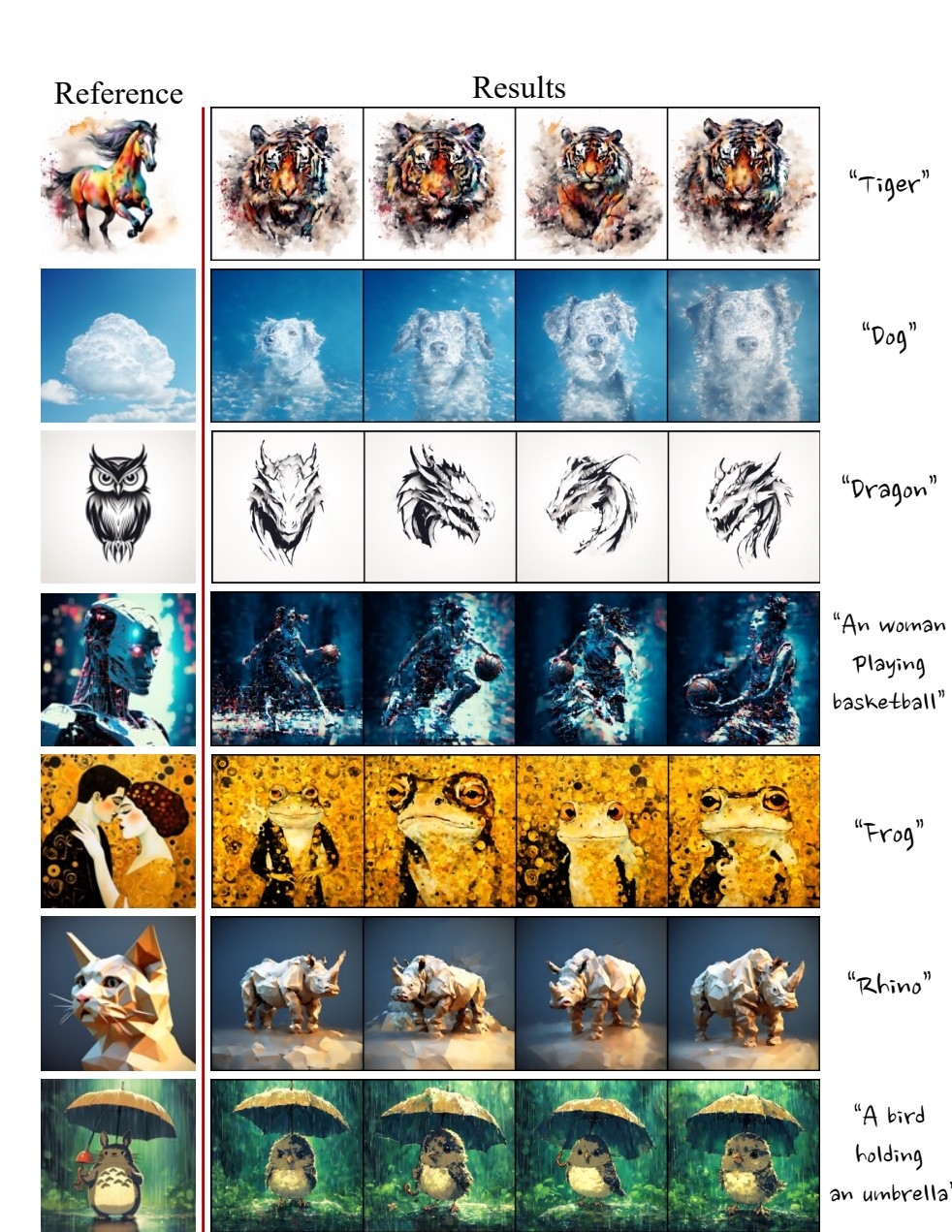

Figure A22: Qualitative result of visual style prompting in Pixart-$\alpha$.

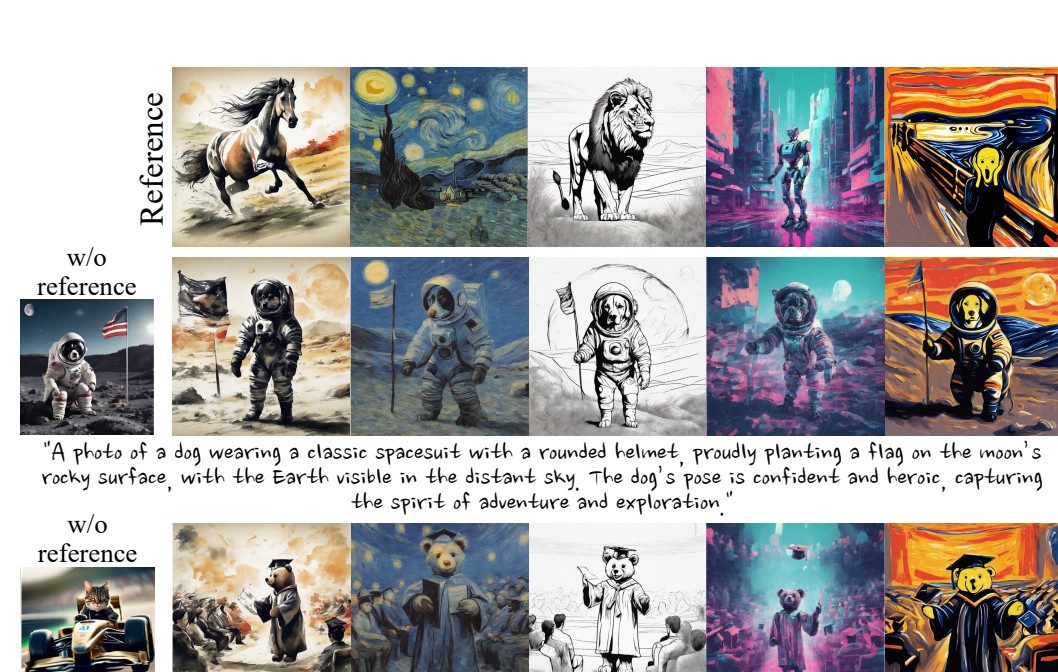

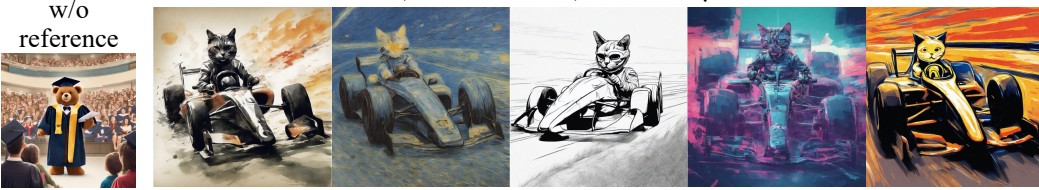

"A photo of a dog wearing a classic spacesuit with a rounded helmet, proudly planting a flag on the moon's rocky surface, with the Earth visible in the distant sky. The dog's pose is confident and heroic, capturing the spirit of adventure and exploration."

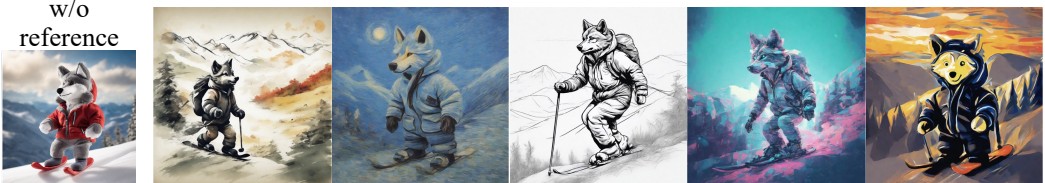

"A photo of a teddy bear dressed in a traditional graduation gown, proudly receiving a diploma on stage, surrounded by an audience of onlookers. The bear's joyful expression reflects the sense of achievement and the importance of this special moment."

"A photo of a cat wearing a sleek racing suit, expertly driving a Formula I car on a track. The cat's focused expression and firm grip on the steering wheel capture the intensity and excitement of high-speed racing."

"A photo of a wolf plush wearing a cozy snowsuit, skillfully skiing down a steep, snow-covered mountain, standing firmly on two legs as it navigates the slope. The wolf's determined posture and the snowy landscape emphasize the thrill and adventure of the downhill journey."

Figure A23: Qualitative result of visual style prompting in complex text prompts.

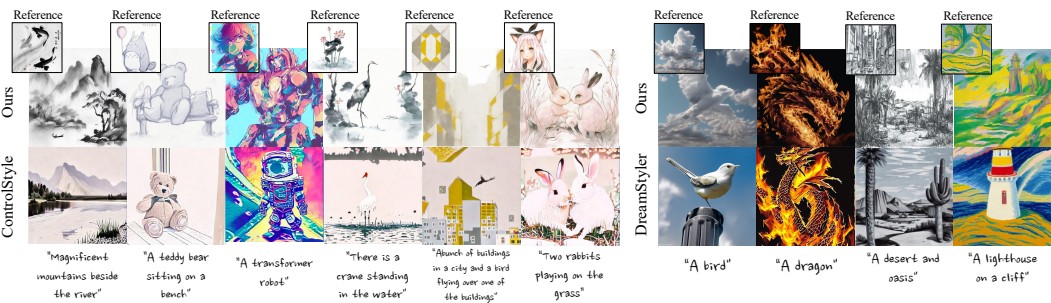

Figure A24: Comparison of ours with DreamStyler and ControlStyle.

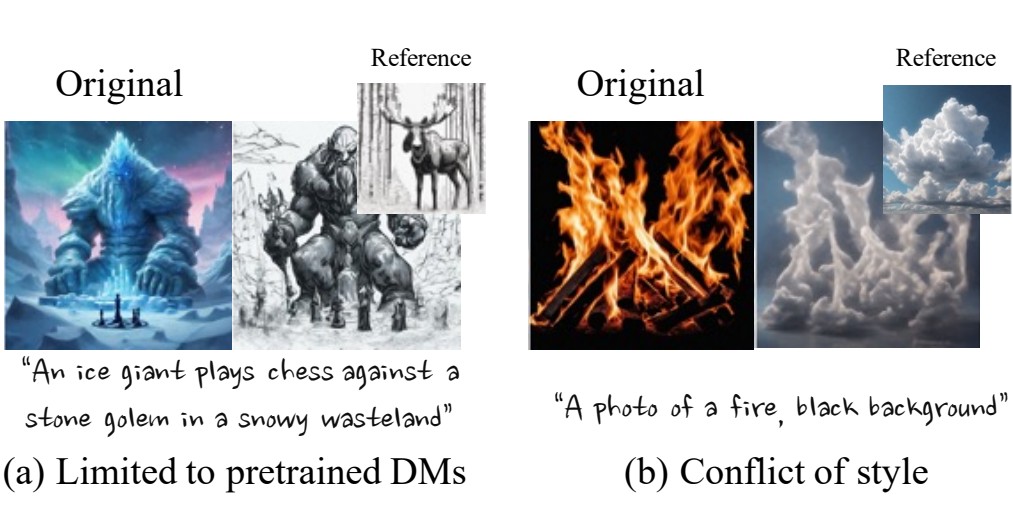

(a) Limited to pretrained DMs        (b) Conflict of style

Figure A25: **Limitation: (1) Text Alignment Failure**: Missing 'stone golem' from the prompt (a), and **(2) Prompt Conflict**: 'Cloud in the sky' vs. 'fire, black background' (b).