# OpenReview forum: "StyleGuide: Crafting visual style prompting with negative visual query guidance"
_ICLR.cc/2025/Conference — Submitted to ICLR 2025_

### Official Review · Reviewer_2yyd · 2024-10-31

**Soundness:** 3
**Presentation:** 3
**Contribution:** 3
**Rating:** 6
**Confidence:** 4

**Summary:**

This paper introduces innovative methods in text-to-image generation, addressing content leakage issues in existing approaches. The authors extend classifier-free guidance (CFG) with swapping self-attention and propose negative visual query guidance (NVQG) to reduce unwanted content transfer. These methods are simple yet effective, achieving precise control over style and content. Extensive evaluations demonstrate the superiority of the proposed methods, ensuring generated images reflect the reference style and match text prompts. Overall, the paper presents significant improvements, providing a solid foundation for future work and practical applications.

**Strengths:**

- The paper is logically structured, providing a thorough analysis of the content leakage issue in style transfer. It proposes the NVQG method to address this problem, thereby improving the quality of generated images.
- The experiments are detailed, with extensive comparative experiments and visual analyses supporting the main contributions of the paper.

**Weaknesses:**

- The proposed CFG with swapping self-attention and  NVQG  in the paper mainly combines previous work, which shows a slight lack of innovation. Additionally, the derivation of equations in section 2.2 is not sufficiently clear.
- Writing and structure: inconsistent terminology usage, NVQG in introduction and NVG in section 3; the layout of Figures 3 and 4 is not aligned; a large number of instances in the formulas where K and V are combined, making it difficult to understand.
-  Experiments comparision on I2I task did not include some of the latest methods of style transfer, such as InstantStyle, InstantStyle-Plus.

**Questions:**

- In the experiments regarding content leakage, the comparison with other models mainly involves qualitative analysis through visualization of certain examples (Figure 7, 10). However, in fact, this paper uses quantitative metrics in Figure 5 to evaluate content leakage. I am curious why these metrics were not used to assess content leakage across different models, considering that content leakage is the main issue this paper aims to address.

---

> ### Author Response · Authors · 2024-11-22
>
> We thank `2yyd` for the thoughtful assessment and the constructive feedback. We carefully address the concerns below.
>
> ## Innovation of StyleGuide
> Thank you for raising your concern. However, we emphasize that while our method builds upon prior research, it highlights previously overlooked issues and proposes a novel approach to address them.
>
> To the best of our knowledge, our research is *the first to propose employing the noise from the style image as the negative guidance*, to combat the content leakage.
>
> Although some studies have explored swapping the key and value of self-attention for stylization (e.g., StyleAligned, CrossAttn, StyleID), none have applied this approach to negative guidance. Please note that negative guidance has traditionally relied on negative text prompts (e.g.,  `"ugly, low-resolution"`) or embeddings trained from such prompts.
>
> Additionally, previous studies have not thoroughly analyzed which layer is most effective for key-value swapping. We conducted ablation experiments and provided a detailed analysis to demonstrate how crucial the selection of the swapping layer is for style prompting.
>
> We also addressed the issues of numerical errors caused by the inversion and color discrepancies observed when using real images as style prompts—problems that were overlooked in prior research. We proposed a simple yet efficient solution using color calibration and stochastic encoding.
>
> ## Derivation of Eq. 2 and terminology
> Thank you for the supportive comments on the derivation.
>
> We will correct the derivation and terminology. We promise to include the full derivation in the appendix until the final submission.
>
> ## Unaligned layout
> Thank you for the supportive feedback. We have fixed the layout in the revised PDF version.
>
> ## More I2I comparison
> We have added comparison results in **Figure 12**.
> Ours consistently balances content preservation and style transfer, accurately reflecting the reference style while maintaining structural integrity across all examples.
>
> [CrossAttn] overemphasizes the style, causing the structure to break down and lose the original content.
>
> [StyleID] reflects the colors of the reference rather than the style itself. Similarly, InstantStyle primarily captures color while slightly distorting the structure. [InstantStyle, InstantStyle+] maintains the structure well but shows inconsistent style reflection depending on the reference image.
>
> ## Content leakage across different models $\rightarrow{}$ L461-472
>  We provide quantitative comparison results with content leakage and style similarity in **Figure A12**.
> It supports that our method shows the best style reflection while not suffering content leakage. We have included them in the revised PDF version.

---

> > ### Comment · Reviewer_2yyd · 2024-11-25
> >
> > Your response addressed my main issue, and I will increase my rating score.

---

> > > ### Author Response · Authors · 2024-11-28
> > >
> > > I’m glad to hear that your concerns have been resolved. Your discussion and effort have been incredibly helpful in improving our paper, and we truly appreciate your valuable contribution.

---

### Official Review · Reviewer_RwJ8 · 2024-11-01

**Soundness:** 3
**Presentation:** 3
**Contribution:** 3
**Rating:** 5
**Confidence:** 3

**Summary:**

This paper propose visual style prompting which receives a text prompt and a visual style prompt to generate new images. Specificaly, this method utilize classifier-free guidance conbined with swapping self-attention to achieve style transfer, and use negative visual query guidance (NVQG) to reduce the transfer of unwanted contents. Extensive experimental verification on both T2I  and I2I has validated the effectiveness of this method.

**Strengths:**

- This paper is well written and easy to follow

- This method is training-free and achieves outstanding preference for style transfer without content leakage

- The experiments and analysis are thoroughly reasonable and justified

**Weaknesses:**

- The methods employed for comparison by the author appear somewhat outdated. Stylization is a rapidly evolving field, as demonstrated by the recent emergence of models such as DEADiff, InstantStyle(-Plus), and CSGO this year. To validate the effectiveness of the proposed approach and assess the issue of content leakage, it is essential to compare it comprehensively with these state-of-the-art techniques.

- Furthermore, the computational efficiency of the algorithm has not been sufficiently analyzed. Given that the method involves multiple attention computations among latent variables, a fair comparison of runtime and memory usage with other methods is essential to assess the feasibility of the proposed approach.

**Questions:**

- In the recently mentioned approaches (such as DEADiff, InstantStyle(-Plus), and CSGO), I haven’t noticed any significant content leakage. Have you verified whether this issue occurs in those methods as well? A direct comparison with these models would help emphasize the strengths of your own approach.

- I also recommend incorporating a few quantitative metrics to evaluate the effectiveness of style transfer. While you don’t need to include many metrics, incorporating some quantitative ones is necessary to validate the effectiveness of the method objectively. Relying solely on selected images can be misleading, as they may have been cherry-picked to showcase the best results.

- Additionally, further experimental analysis on computational efficiency is advised to provide a more comprehensive evaluation of the method.

I will revise my rating according to the author's feedback and the reviewer's discussion.

**Details Of Ethics Concerns:**

This paper has no ethical concerns.

---

> ### Author Response · Authors · 2024-11-22
>
> We thank `RwJ8` for the thoughtful assessment and the constructive feedback. We carefully address the concerns below.
>
> ## Additional Competitors
>
> Thank you for helping us enhance the rigor of our analysis. We provide qualitative and quantitative comparison results with [DEADiff, InstantStyle, and CSGO] in **Figure A11** and **Figure A12**, respectively.
>
> Our method achieves superior style reflection without experiencing content leakage, producing detailed, coherent, and visually compelling results.
>
> In contrast, [DEADiff, InstantStyle, and CSGO] *fail to adequately reflect style elements* such as texture, object color, background color, and material in references like `A fire` and `A cloud`.
>
> More specifically, the results from [DEADiff] exhibit inconsistent styles within the same reference image (e.g., results for `The Kiss` painting show variations in background color, patterns, and painting style). Similarly, [InstantStyle] generates background patterns and painting styles not present in the reference image (e.g., `The Kiss` painting or `A helmet` reference). [CSGO], on the other hand, produces artifacts with repetitive patterns absent in the reference image.
>
>
>
> ## Computational efficiency
> Thank you for raising your concern.
> Inference time is not an obstacle to our method because it takes only `(N+1)/N` times of the vanilla generation for sampling `N` images. `+1` in the numerator is the style reference.
>
> For sampling 6 images, the table below shows the inference time and memory usage of SDXL and SDXL with our method.
>
>
>
> |                      | Inference time (seconds) | Memory usage (GB) |
> |:--------------------:|:------------------------:|:-----------------:|
> |         SDXL         |           163            |      20.131       |
> | SDXL with our method |           191            |      22.169       |
>
>
> ## Content leakage in the other methods
> We appreciate you bringing up this concern.
>
> Although we agree that [DEADiff] and [InstantStyle(-Plus)] effectively avoid content leakage, they fall short in achieving sufficient style reflection, as shown in **Figure A11**. Avoiding content leakage through insufficient stylization may be effective, but it undermines the essence of stylization and is therefore undesirable. In contrast, our method achieves optimal stylization without content leakage.
>
>
> ## Incorporating quantitative metrics for style transfer
>
> We appreciate your attention to the quantitative metrics used to evaluate the effectiveness of style transfer.
>
> Following [P+, Dreambooth], we provide quantitative results using DINO similarity in **Figure 8**. Additionally, we have included quantitative results based on Gram loss, as proposed by [Gatys], in **Figure A12**.
>
> As shown in the table, our method achieves the best style similarity while maintaining text alignment.
>
> [P+]: Extended textual conditioning in text-to-image generation
>
> [Dreambooth]: Fine tuning text-to-image diffusion models for subject-driven generation, cvpr 2023
>
> [Gatys]: A Neural Algorithm of Artistic Style

---

> > ### Comment · Reviewer_RwJ8 · 2024-11-27
> > **Official Comment by Reviewer RwJ8**
> >
> > Thank you for your response. After reading all the replies, I plan to keep my score.

---

> > > ### Author Response · Authors · 2024-11-27
> > >
> > > Thank you for your thoughtful review and for considering our responses. We truly appreciate your efforts.
> > >
> > > We understand your decision to maintain your score. However, we believe we have addressed all your concerns and clarified any uncertainties raised earlier.
> > >
> > > If there are any remaining issues or areas where we can further demonstrate the contributions of our work, please let us know.
> > >
> > > We remain committed to improving and ensuring the clarity of our research.
> > > Thank you again for your valuable feedback.

---

### Official Review · Reviewer_WuwM · 2024-11-04

**Soundness:** 3
**Presentation:** 2
**Contribution:** 2
**Rating:** 6
**Confidence:** 4

**Summary:**

This paper focuses on Style Transfer using diffusion-based models, in which a style given by a reference image is transferred to the input image (e.g., transferring the "polygon" style in the reference image to a photo of a dog).
To that end, this paper conducts an extensive study on each component of diffusion-based models (e.g., classifier-free guidance, negative prompts, etc.) to propose a final approach that achieves state-of-the-art performance in style transfer.

**Strengths:**

- Overall, style transfer is a popular topic in the Image Editing community with many useful applications.
- This paper carries out thorough studies and proposes a method to prevent content leakage based on the insights. I think both the insights and the proposed method are useful to readers, as "content leakage" is a common problem in many image editing/manipulation methods.

**Weaknesses:**

I find it unclear how competitive the proposed method is compared to existing work (e.g., StyleDrop). I also wonder if inference time might limit this approach, as it involves multiple steps. Furthermore, with each new {reference style, input image} pair, all these steps need to be repeated.

**Questions:**

I’m inclined to accept this paper due to its superior quantitative results, it'd be much appreciated if authors can clarify in the rebuttal about training/ inference of proposed methods vs. existing works.

---

> ### Author Response · Authors · 2024-11-22
>
> We thank `WuwM` for the positive assessment and the constructive feedback. We carefully address the concerns below.
> ## Comparison with StyleDrop
> Fortunately, we have the comparison with StyleDrop in the paper (**Figure 7**).\
> Our method better reflects style elements from the reference. Please understand that we used the unofficial StyleDrop repo because StyleDrop does not open-source the official code.
>
> ## Inference time and training requirement
> Thank you for raising your concern.
> Inference time is not an obstacle to our method because it takes only  `(N+1)/N ` times of the vanilla generation for sampling  `N ` images.  `+1 ` in the numerator is the style reference.
>
> Our approach *does not require additional training*. It leverages pre-trained models and works directly during inference, ensuring computational efficiency and practicality without extra training overhead.
>
> For sampling 6 images, the table below shows the inference time and memory usage of SDXL and SDXL with our method.
>
>
> |                      | Inference time (seconds) | Memory usage (GB) |
> |:--------------------:|:------------------------:|:-----------------:|
> |         SDXL         |           163            |      20.131       |
> | SDXL with our method |           191            |      22.169       |

---

### Official Review · Reviewer_shSn · 2024-11-05

**Soundness:** 3
**Presentation:** 3
**Contribution:** 3
**Rating:** 8
**Confidence:** 4

**Summary:**

This paper sets the new state-of-the-art method for image generation tasks given visual style prompts. This paper mainly addresses the content leakage issue by incorporating swapping self-attention in CFG and utilizing negative visual guidance. More specifically, the improved CFG ensures the balance between style and content, and using negative visual guidance suppresses the content leaking into the generated image. Further, stochastic encoding and color calibration tricks are introduced to improve the generation quality.

**Strengths:**

- The quantitative and qualitative assessments showcase the superior effectiveness of the method proposed.

- The intuitive and effective negative visual guidance proposed serves to prevent content leakage.

- Comprehensive experiments are carried out to aid readers in understanding the proposed pipeline and key components.

**Weaknesses:**

- Certain design decisions (such as determining the optimal layers for balancing style and content, exchanging self-attention, and color calibration) exhibit effectiveness but lack in-depth analysis or theoretical derivation.

- I would appreciate seeing quantitative ablation studies to further illustrate the effectiveness of stochastic encoding and color calibration. Do they play the most crucial role in the end outcomes? If that is the case, the effectiveness of self-attention swapping in CFG and the use of negative visual guidance diminishes. Furthermore, since they could potentially be integrated into other diffusion-based techniques, exploring their utility in other methods would be interesting.

- Typo correction: Line 18 employ -> employs.

**Questions:**

It would be great if the weaknesses raised above could be addressed in the rebuttal.

---

> ### Author Response · Authors · 2024-11-22
>
> We thank `shSn` for the positive assessment and the constructive feedback. We carefully address the concerns below.
>
> ## In-depth analysis & theoretical derivation for design decisions
> While we selected the model design in a sophisticated manner based on experiments and analyses from existing papers, we acknowledge that some design choices lacked in-depth analysis and explanation.
> We sincerely appreciate the constructive review and intend to incorporate the following details into the paper.
>
> ### (1) Optimal layers for balancing style and content $\rightarrow$ L230-247, L339-355
>
> - We disregard the bottleneck feature because it is known to represent content and attributes [Asyrp,InjectFusion, park2024].
> - We disregard the downblocks because the self-attention features do not form a clear layout and structure at the downblocks  (**Figure 4** in [MasaCtrl, meng2024]).
>
> - We choose the late upblocks rather than the early upblocks because swapping self-attention attends to the style correspondence more at the late upblocks than the early upblocks. This analysis is provided in **Figure 6**.
>
> [Asyrp]: Diffusion Models Already Have A Semantic Latent Space, iclr 2023
>
> [InjectFusion]: Training-free Content Injection using h-space in Diffusion models, wacv 2024
>
> [park2024]: Understanding the latent space of diffusion models through the lens of riemannian geometry, neurips 2023
>
> [MasaCtrl]: Tuning-free mutual self-attention control for consistent image synthesis and editing, iccv 2023
>
> [meng2024]: Not All Diffusion Model Activations Have Been Evaluated as Discriminative Features, neurips 2025
>
>
> ### (2) Exchanging self-attention $\rightarrow$ L101-137
> Self-attention layers use spatial dimensions (height × width) to represent visual elements, while cross-attention layers use non-spatial tokens (text token length). To reflect style elements from a reference image that are difficult to capture textually, we borrow keys and values from self-attention layers during the reference process, a method we term swapping self-attention.
> In addition, swapping self-attention has a strong connection with style transfer literature [aams, sanet, mast, adaattn, styletr2] where the attention mechanism reassembles visual features of a style image (key, value) on a content image (query).
> Instead of a content image, our method has a random noise and a text prompt for specifying the content.
>
> [aams] Attention-aware multi-stroke style transfer, Yao+, cvpr 2019
>
> [sanet] Arbitrary Style Transfer with Style-Attentional Networks, Park and Lee, cvpr 2019
>
> [mast] ArFbitrary style transfer via multi-adaptation network, Deng+, acmmm 2020
>
> [adaattn] Adaattn: Revisit attention mechanism in arbitrary neural style transfer, Liu+, iccv 2021
>
> [styletr2] Stytr2: Image style transfer with transformers, Deng+, cvpr 2022
>
> ### (3) Color calibration $\rightarrow$ L264-294
> Color calibration builds on the general knowledge that images within the same style have similar channel-wise statistics [Gatys]. However, the noisy denoising process makes it difficult to match statistics of the noisy latent to the target statistics. To address this, we proposed to use predicted x0.
>
> [Gatys]: A Neural Algorithm of Artistic Style
>
> ## Quantitative ablation studies of stochastic encoding and color calibration
>
> Thank you for raising your concern.
>
> We would like to clarify that the effectiveness of stochastic encoding does not diminish the effectiveness of the other methods, as stochastic encoding addresses an independent problem: `how to use a real image as a reference`. If we use a generated image as a reference, stochastic encoding is not required.
> We provide a quantitative ablation study of each configuration in ablation **Figure A19**. As shown in the figure, swapping self-attention and employing NVQG improve performance (style reflection & text alignment) regardless of whether the reference image is real or generated.
>
> On the other hand, stochastic encoding competes with DDIM inversion and demonstrates improved performance.
>
> Lastly, we note that there is no priority in terms of importance among the four proposed methods: swapping self-attention, NVQG, stochastic encoding, and color calibration.
>
> We have added them in the revised PDF L485-502.
>
> ### Typo
> Thank you for pointing that out. We have corrected the typo in the revised PDF.

---

> > ### Comment · Reviewer_shSn · 2024-11-28
> >
> > I appreciate the authors' efforts to address my concerns. The analysis, i.e., discussions of other papers regarding the design choices provides more helpful information for readers to grasp the pipeline. Plus, the new ablation study in Figure A19 resolves my concern. Therefore, I still recommend accepting this paper.

---

> > > ### Author Response · Authors · 2024-11-28
> > >
> > > Thank you for your positive feedback and for taking the time to review our revisions. We are glad to hear that the additional analysis and ablation study addressed your concerns and provided more clarity on our design choices. Your insightful comments have been invaluable in improving the quality and presentation of our work.
> > >
> > > We sincerely appreciate your recommendation and support.

---

### Author Response · Authors · 2024-11-27

As the PDF update period is ending soon, we kindly remind the missing reviewers in the discussion.

We appreciate all the reviewers for acknowledging our strengths:
- The superior effectiveness of the proposed method compared to previous methods is supported by the quantitative and qualitative results.
- Negative visual query guidance intuitively and effectively prevents content leakage.
- Comprehensive experiments of the key components aid the reader's understanding.
- The target task is useful and popular.
- The paper is well-written and easy to follow.
- The proposed method is **training-free**.
- The users highly prefer their results.
- The experiments and analysis are thoroughly reasonable and justified.

And providing ingredients to strengthen our paper:
- In-depth analysis or theoretical derivation of design decisions.
- Quantitative ablation studies of stochastic encoding and color calibration.
- More comparison with existing works (e.g., DEADiff, CSGO, InstantStyle(-Plus))
- Inference time & memory usage
- Typo correction
- Content leakage of the other methods
- More quantitative metrics to measure style similarity


In the rebuttal, we have carefully addressed the comments so that the reviewer can anticipate our high-quality camera-ready version. If there are any other questions or concerns, please feel free to post another comment.

---

### Meta-Review · Area_Chair_L8qt · 2024-12-21

**Metareview:**

The paper addresses the problem of styled image generation using a reference style image, specifically focusing on the issue of content leakage, where unintended content from the style prompt is transferred to the generated output. To solve this problem, the authors propose StyleGuide, which leverages attention swapping and negative visual guidance to reduce content leakage during generation.

Strength
* The target problem has many real-world applications
* Extensive experiments demonstrate the effectiveness of the proposed method in styled image generation
* The paper offers detailed analysis which provide useful insight for the content leakage problem in diffusion model

Weakness
* The design choices for the approach lack deeper theoretical justification or thorough ablation studies
*  The paper does not thoroughly analyze the computational complexity of the proposed method, raising concerns about fairness in comparisons.
* The baselines considered in the experiments are outdated, while newer methods do not exhibit the content leakage problem targeted in this paper. This undermines the relevance and necessity of the proposed approach.
* The method builds on top of prior work by combining existing techniques, and the technical novelty and contribution is incremental

The reviewers acknowledge the practical importance of the problem and note the promising experimental results. However, they express concerns about the paper’s technical novelty, the relevance of the content leakage problem, and the lack of comparisons with state-of-the-art methods. In particular, Reviewer RwJ8 notes that content leakage is not evident in recent style image generation methods, raising question about the paper’s core claim and contribution. While the paper offers a reasonable approach, the limited novelty and the lack of solid justification for the problem’s relevance make the contribution borderline. Additional justification is needed to secure the paper’s impact and significance.

**Additional Comments On Reviewer Discussion:**

The authors addressed concerns about design choices and computational complexity by providing additional explanations and results in the rebuttal. They also included comparisons with additional baselines; however, the experiments remain insufficient to fully justify the contribution.

---

### Decision · Program_Chairs · 2025-01-22

Reject